# IRS-1 acts as an endocytic regulator of IGF-I receptor to facilitate sustained IGF signaling

Yosuke Yoneyama[1], Peter Lanzerstorfer[2], Hideaki Niwa[3,4], Takashi Umehara[3,4,5], Takashi Shibano[1†], Shigeyuki Yokoyama[3,6], Kazuhiro Chida[1], Julian Weghuber[2,7], Fumihiko Hakuno[1]*, Shin-Ichiro Takahashi[1]*

[1]Department of Animal Resource Sciences, Graduate School of Agriculture and Life Sciences, The University of Tokyo, Tokyo, Japan; [2]University of Applied Sciences Upper Austria, Wels, Austria; [3]RIKEN Systems and Structural Biology Center, Yokohama, Japan; [4]RIKEN Center for Life Science Technologies, Yokohama, Japan; [5]PRESTO, Japan Science and Technology Agency, Kawaguchi, Japan; [6]RIKEN Structural Biology Laboratory, Yokohama, Japan; [7]Austrian Competence Center for Feed and Food Quality, Safety and Innovation, Wels, Austria

*For correspondence:
ahakuno@mail.ecc.u-tokyo.ac.jp (FH);
atkshin@mail.ecc.u-tokyo.ac.jp (S-IT)

Present address: †Department of Oncology and Pathology, Karolinska Institutet and Karolinska University Hospital, Stockholm, Sweden

Competing interests: The authors declare that no competing interests exist.

**Abstract** Insulin-like growth factor-I receptor (IGF-IR) preferentially regulates the long-term IGF activities including growth and metabolism. Kinetics of ligand-dependent IGF-IR endocytosis determines how IGF induces such downstream signaling outputs. Here, we find that the insulin receptor substrate (IRS)−1 modulates how long ligand-activated IGF-IR remains at the cell surface before undergoing endocytosis in mammalian cells. IRS-1 interacts with the clathrin adaptor complex AP2. IRS-1, but not an AP2-binding-deficient mutant, delays AP2-mediated IGF-IR endocytosis after the ligand stimulation. Mechanistically, IRS-1 inhibits the recruitment of IGF-IR into clathrin-coated structures; for this reason, IGF-IR avoids rapid endocytosis and prolongs its activity on the cell surface. Accelerating IGF-IR endocytosis via IRS-1 depletion induces the shift from sustained to transient Akt activation and augments FoxO-mediated transcription. Our study establishes a new role for IRS-1 as an endocytic regulator of IGF-IR that ensures sustained IGF bioactivity, independent of its classic role as an adaptor in IGF-IR signaling.
DOI: https://doi.org/10.7554/eLife.32893.001

## Introduction

Insulin-like growth factor (IGF)-I receptor (IGF-IR) is an important receptor tyrosine kinase (RTK) that regulates a variety of biological processes including proliferation, cell survival, and control of metabolism in a wide range of mammalian tissues by binding the ligands IGF-I and IGF-II (*Nakae et al., 2001*). Ligand binding to the IGF-IR extracellular domain causes conformational changes of the intracellular region, inducing the tyrosine kinase domain to autophosphorylate multiple Tyr residues and activate intrinsic RTK activity (*Kavran et al., 2014*; *Favelyukis et al., 2001*). IGF-IR then initiates downstream signaling through tyrosine phosphorylation of insulin receptor substrate (IRS) adaptor proteins to activate the phosphatidylinositol 3-kinase (PI3K)-Akt pathway and its various biological responses (*Myers et al., 1996*; *Sun et al., 1993*; *White, 2002*).

IGF/IGF-IR stimulates the PI3K-Akt pathway in a stereotypical way – sustained tonal induction. Sustained induction is thought to define the specific biological outcomes of IGF signaling, and distinguish the function of the IGF ligand from other RTKs/ligands that access the Akt cascade (*Gross and Rotwein, 2016*; *Kubota et al., 2012*). In particular, sustained activation of the PI3K-Akt pathway, mediated by IGF-IR, induces cell proliferation in multiple types of cells, cell survival in neural cells,

**eLife digest** Mammals, including humans, use signaling molecules called hormones to carry information from one cell to another. Insulin-like growth factor (or IGF for short) is a hormone that is essential throughout an animal's lifetime. It is needed for growth and for many of the chemical processes that must occur to maintain life (which are collectively referred to as an animal's metabolism). IGF binds to and activates a protein found on the surface of cells, which then transmits the signal inside the cells. This surface protein is known as the IGF-I receptor, and once it is activated by IGF binding, it is removed from the cell surface and then incorporated inside the cell to switch off the signal. The IGF signal in cells needs to be properly balanced to prevent disorders of growth and metabolism.

How long the activated IGF-I receptor remains at the cell surface and when the IGF-I receptor starts to enter inside the cells after cells receive IGF influence the signals within the cell. Often IGF signaling must be activated for long periods, for example when cells maintain their balance between making and breaking proteins. However, it remains poorly understood how the IGF-I receptor produces a sustained signal.

Yoneyama et al. have now focused on a protein called IRS-1, which was known to act downstream of the receptor. The experiments revealed that this protein determines how long activated IGF-IR remains at the cell surface before it enters inside cells. It achieves this by binding to a complex of proteins, known as AP2, which normally internalizes the IGF-I receptor. However, when IRS-1 binds, it inhibits AP2. This means that the receptor is no longer rapidly removed from the cell surface and can continue signaling for long periods of time.

The findings of Yoneyama et al. help to explain how long-term IGF signaling is regulated. Further work that builds on these findings could help scientists to understand how uncontrolled IGF signals cause the development of diseases including cancer and metabolic disorders.

DOI: https://doi.org/10.7554/eLife.32893.002

and protein homeostasis in skeletal muscle cells (*Fernandez and Torres-Alemán, 2012*; *Fukushima et al., 2012*; *Ness and Wood, 2002*; *Sacheck et al., 2004*; *Stewart and Rotwein, 1996*). To date, the mechanism by which IGF-IR produces sustained signaling remains poorly understood.

Clathrin-mediated endocytosis (CME) is a major regulator of RTKs (*Goh and Sorkin, 2013*) involving the heterotetrameric AP2 complex composed of large α and β2, medium μ2, and small σ2 subunits (*Collins et al., 2002*). AP2 binds to transmembrane cargo proteins that contain specific motifs such as YxxΦ (Y denotes Tyr; x, any amino acid; and Φ, bulky hydrophobic residue) serving as μ2 binding sites (*Owen and Evans, 1998*; *Traub and Bonifacino, 2013*). In addition, AP2 associates with clathrin and with endocytic accessory proteins at the plasma membrane to coordinate clathrin-coated pit (CCP) formation (*Schmid and McMahon, 2007*). Ligand-bound RTKs enter the endocytic process through CME, but perhaps with different signaling consequences. If endocytosed RTKs are sorted to lysosomes for degradation, this process down-regulates signaling as exemplified by the model RTKs including epidermal growth factor receptor (EGFR) and platelet-derived growth factor receptor (*Goh and Sorkin, 2013*). On the other hand, some RTKs continue to signal locally across the endosome membrane even after endocytosis (*Schenck et al., 2008*; *Villaseñor et al., 2015*; *Lin et al., 2006*). In either case, RTK internalization strongly impacts its signaling outputs. Thus, the duration at the cell surface of ligand-bound RTKs, which is tightly regulated by CME, critically fine-tunes their signaling and biological functions. Accordingly, we hypothesize that ligand-bound IGF-IR, which exhibits sustained activation and slow degradation (*Fukushima et al., 2012*; *Mao et al., 2011*; *Zheng et al., 2012*), undergoes slow or delayed CME (*Martins et al., 2011*; *Monami et al., 2008*). To evaluate this idea, here we study the molecular components regulating perdurance of IGF-IR at the cell surface through its interactions with CME, and elucidate how this dictates IGF signaling and outputs.

Among IRS family proteins IRS-1 and IRS-2 are well known as major substrates of IGF-IR (*Taniguchi et al., 2006*; *White, 2002*). We and others have shown that IRS-associated proteins contribute to the regulation of IRS-1/IRS-2 function through distinct mechanisms (*Ando et al., 2015*;

*Hakuno et al., 2015, 2007; Lee et al., 2013; Shi et al., 2011; Fukushima et al., 2015; Yoneyama et al., 2013*). In this study, we discovered AP2 is also an IRS-1-associated protein. Unexpectedly IRS-1 promotes the surface retention of activated IGF-IR through inhibiting AP2-dependent internalization of IGF-IR, and this is independent of IRS's classic role as an adaptor protein in IGF-IR and insulin receptor signaling. The ability of IRS-1 to prolong surface retention of IGF-IR is essential for long-term PI3K-Akt signaling. Our results establish a novel role of IRS-1 in ensuring the sustained effects of IGFs via its direct control of IGF-IR internalization.

## Results

### IRS-1 interacts with the clathrin adaptor AP2 complex through its Yxx$\varphi$ motifs

To identify the IRS-1-interacting proteins that potentially regulate insulin/IGF signaling, we searched the candidates in our previous yeast two-hybrid screening (*Hakuno et al., 2007*). We found the μ2 subunit of clathrin adaptor AP2 complex among the frequently obtained clones (*Figure 1A*). Co-immunoprecipitation assay using HEK293T cells expressing FLAG-tagged IRS-1 or IRS-2 revealed that endogenous AP2 subunits (α-adaptin and μ2) were detected in a complex with IRS-1, but not with IRS-2 (*Figure 1B*). In addition, a portion of AP2 was co-immunoprecipitated with endogenous IRS-1 as well as ectopically expressed FLAG-IRS-1 in L6 myoblasts, and this interaction was not affected by IGF-I stimulation (*Figure 1C,D*).

Using IRS-1 truncated mutants, we mapped the central region (amino acid residues 543–865) which is necessary for the binding to AP2 (*Figure 1E*). This region is almost identical to that for the clathrin adaptor AP1 complex found in our previous study, which binds to YxxΦ motifs of IRS-1 including Tyr 608, Tyr 628, and Tyr 658 via its μ1 subunit (*Yoneyama et al., 2013*). Indeed, the Ala mutation of all these Tyr residues in IRS-1, but not a single substitution, completely abolished the binding to μ2 in vitro (IRS-1 3YA mutant; *Figure 1F,G*). We also analyzed the crystal structures of μ2 C-terminal subdomain (C-μ2) bound to IRS-1 YxxΦ motifs (*Figure 1—figure supplement 1A,B* and *Table 1*). Importantly, the side chains of Tyr and Met residues of IRS-1 YxxΦ motifs are inserted into the binding pockets of μ2, which are shared by the AP2 cargo proteins (*Owen and Evans, 1998*) (*Figure 1—figure supplement 1C*). Collectively, these results indicate that IRS-1 is recognized by the AP2 complex via the μ2 subunit in the very similar manner to conventional endocytic cargos.

The μ2 subunit of AP2 cannot recognize phosphorylated YxxΦ sequence due to its limited capacity (*Kittler et al., 2008; Owen and Evans, 1998*). However, IGF-I stimulation did not inhibit the co-immunoprecipitation of IRS-1 with AP2 (*Figure 1C,D*). To evaluate the stoichiometry of IRS-1 Tyr phosphorylation in IGF-I-stimulated cells, we analyzed the amount of IRS-1 capable of binding to GST-C-μ2 in lysates of cells treated with or without IGF-I (*Figure 1—figure supplement 1D,E*). Although the amounts of both pulled-down and immunoprecipitated IRS-1 were comparable, Tyr-phosphorylated IRS-1 was hardly pulled down by μ2 (*Figure 1—figure supplement 1F*), indicating low stoichiometry of IRS-1 Tyr phosphorylation after IGF-I stimulation and existence of a non-phosphorylated IRS-1 pool which interacts with AP2.

### IRS-1 promotes cell surface retention of activated IGF-IR via its Yxx$\varphi$ motifs

Since AP2 plays a central role in the CME of RTKs, we reasoned that the interaction of IRS-1 with AP2 affects the internalization of IGF-IR. Using the surface biotinylation assay, we first analyzed the changes in cell surface IGF-IR in L6 cells. Long-term stimulation with IGF-I (3 to 12 hr) induced the significant reduction of phosphorylated IGF-IR (phospho-IGF-IR), which was assessed by Tyr 1131 phosphorylation in the activation loop (*Favelyukis et al., 2001*), at the cell surface (*Figure 2A,B*). Similar results were obtained in the analyses of other phosphorylation sites in IGF-IR (*Figure 2—figure supplement 1A*). No reduction of phospho-IGF-IR or total IGF-IR at the cell surface was observed during short-term stimulation with IGF-I (5 to 60 min) (*Figure 2A and B*). Ubiquitination of IGF-IR has been proposed as an important event inducing its internalization and down-regulation (*Monami et al., 2008; Mao et al., 2011*). We observed that IGF-I-induced ubiquitination of IGF-IR reached the maximum 60 min after IGF-I stimulation in L6 cells (*Figure 2—figure supplement 1B*).

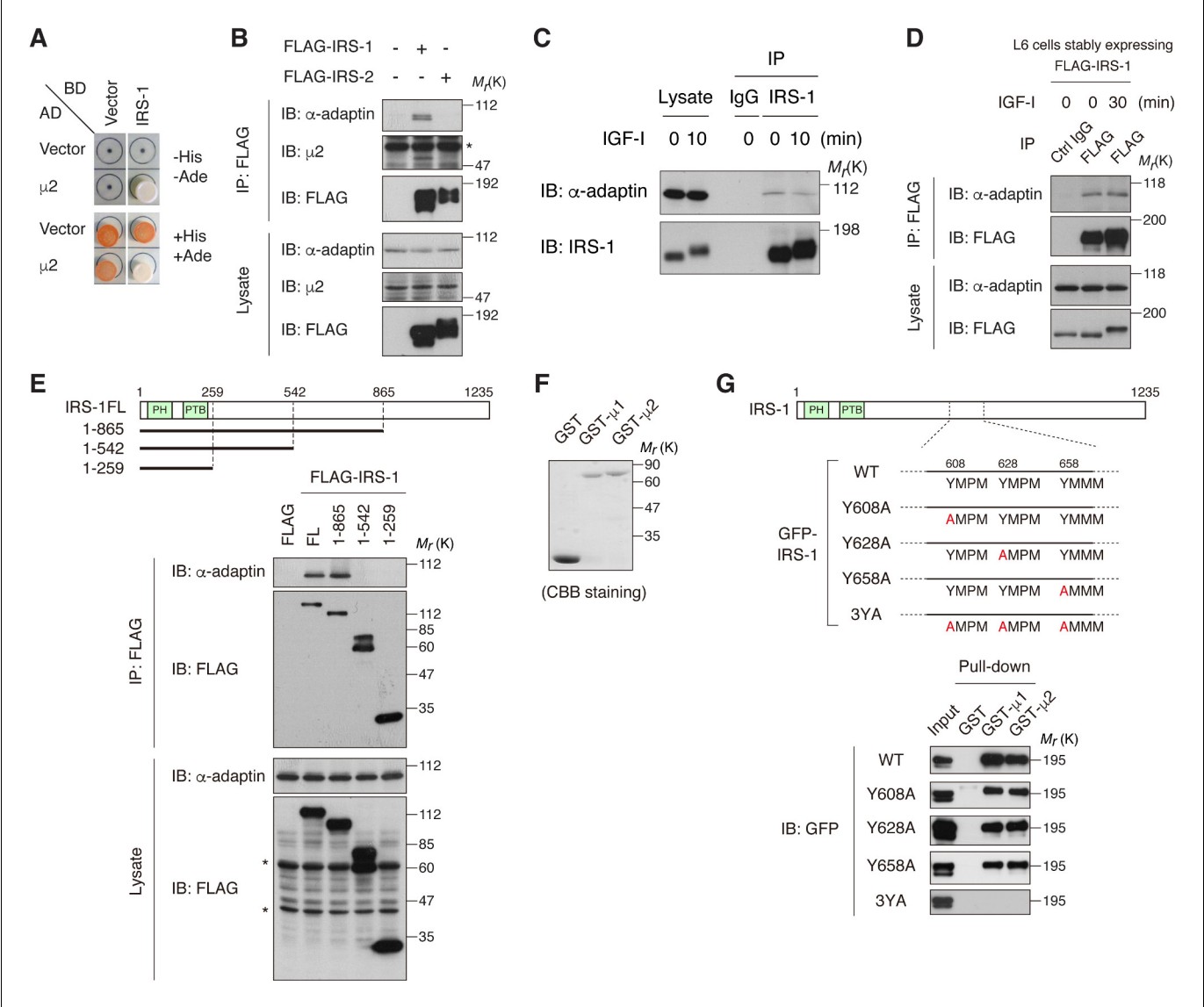

**Figure 1.** IRS-1 interacts with the clathrin adaptor AP2 complex through its YxxΦ motifs. (**A**) Yeast two-hybrid assay indicating the interaction of IRS-1 with the μ2 subunit of AP2. (**B**) The association of IRS-1 or IRS-2 with endogenous AP2 subunits was analyzed by immunoprecipitation in HEK293T cells expressing FLAG-IRS-1 or FLAG-IRS-2. Asterisk indicates IgG band. (**C, D**) Changes in endogenous IRS-1- (**C**) and ectopically expressed FLAG-IRS-1- (**D**) associated AP2 following IGF-I stimulation in L6 cells were analyzed by immunoprecipitation. (**E**) AP2-binding region on IRS-1 was mapped with the indicated truncation mutants of FLAG-IRS-1 by immunoprecipitation of HEK293T cell lysates. Asterisks indicate nonspecific bands. (**F, G**) In vitro pull-down assay for the interaction between IRS-1 mutants and μ2 subunit. Coomassie brilliant blue (CBB) staining of the recombinant proteins (GST, GST-μ1, and GST-μ2) used in the pull-down assay is shown (**F**). Three YxxΦ motifs in IRS-1, which contain Y608, Y628, and Y658 are depicted. The lysates from HEK293T cells expressing the indicated GFP-IRS-1 mutants were pulled down with GST-fused μ1 and μ2 (**G**).
DOI: https://doi.org/10.7554/eLife.32893.003
The following figure supplement is available for figure 1:

**Figure supplement 1.** Three YxxΦ motifs in IRS-1 mediate the interaction with μ2 of AP2 complex.
DOI: https://doi.org/10.7554/eLife.32893.004

We next generated L6 cell lines stably expressing IRS-1 fused with green fluorescent protein (GFP-IRS-1) (*Figure 2C*). Strikingly, phospho-IGF-IR at the cell surface was sustained even after prolonged IGF-I stimulation in GFP-IRS-1-expressing cells while the reduction was observed in the control cells expressing GFP only (*Figure 2D,E*). In contrast, GFP-IRS-2 expression did not affect the reduction in phospho-IGF-IR (*Figure 2—figure supplement 1C,D*). To investigate the requirement

**Table 1.** Data collection and refinement statistics

| | Y608 peptide complex | Y628 peptide complex | Y658 peptide complex |
|---|---|---|---|
| Crystal parameters | | | |
| Space group | $P6_4$ | $P6_4$ | $P6_4$ |
| Cell dimensions: | | | |
| $a$, $b$, $c$ (Å) | 126.07, 126.07, 73.40 | 126.19, 126.19, 74.11 | 125.48, 125.48, 74.14 |
| $\alpha$, $\beta$, $\gamma$ (°) | 90, 90, 120 | 90, 90, 120 | 90, 90, 120 |
| Data collection | | | |
| Wavelength (Å) | 1.000 | 1.000 | 1.000 |
| Resolution (Å) | 50–2.63 (2.68–2.63)* | 50–3.10 (3.15–3.10) | 50–2.60 (2.64–2.60) |
| No. of unique reflections | 20035 | 12419 | 20659 |
| Multiplicity | 11.3 (10.9) | 11.3 (11.4) | 11.4 (11.5) |
| Completeness (%) | 100 (100) | 100 (100) | 100 (100) |
| $R_{meas}$ | 0.078 (1.504) | 0.103 (1.880) | 0.094 (2.069) |
| $R_{pim}$ | 0.023 (0.455) | 0.031 (0.556) | 0.028 (0.608) |
| $CC_{1/2}$ | (0.743) | (0.646) | (0.780) |
| Mean $I/\sigma$ | 28.1 (1.8) | 24.8 (1.6) | 26.5 (1.6) |
| Refinement | | | |
| Resolution (Å) | 43–2.62 | 36–3.10 | 36–2.60 |
| No. of reflections | 19977 | 12322 | 20589 |
| $R_{work}/R_{free}$ | 0.185/0.223 | 0.194/0.251 | 0.192/0.227 |
| RMSD bond lengths (Å) | 0.008 | 0.010 | 0.009 |
| RMSD bond angles (°) | 0.948 | 1.194 | 0.965 |
| No. of atoms | | | |
| Protein/peptide | 2003 | 2121 | 2118 |
| Water/ion | 2 | 0 | 34 |
| Ramachandran plot | | | |
| Favored (%) | 95.5 | 92.3 | 95.4 |
| Outliers (%) | 0 | 0 | 0 |
| PDB accession code: | 5WRK | 5WRL | 5WRM |

*Values in parentheses are for highest resolution shell.

DOI: https://doi.org/10.7554/eLife.32893.005

of IRS-1 interaction with AP2 for the surface retention of phospho-IGF-IR, we analyzed the cells expressing the GFP-IRS-1 3YA mutant, which lacks the binding motifs for the μ2 subunit of AP2 complex. In contrast to GFP-IRS-1 wild-type (WT)-expressing cells, surface phospho-IGF-IR was reduced by prolonged IGF-I stimulation in GFP-IRS-1 3YA-expressing cells (*Figure 2D,E*). These data strongly suggest that IRS-1 can promote cell surface retention of activated IGF-IR via its YxxΦ motifs.

The Tyr residues of the YxxΦ motifs of IRS-1 for binding to AP2 (Tyr 608, Tyr 628, and Tyr 658) are known to be phosphorylated by IR/IGF-IR and in turn serve as putative binding sites of PI3K (*Sun et al., 1993*; *Myers et al., 1996*). We next asked whether their Tyr phosphorylation of IRS-1 is involved in the surface retention of IGF-IR. Here, we used the IRS-1 ΔPTB mutant which lacks the phosphotyrosine binding domain (PTB) and therefore cannot be phosphorylated due to the inability to interact with IGF-IR (*Figure 2—figure supplement 1E*). As with GFP-IRS-1 WT, expression of GFP-IRS-1 ΔPTB resulted in the surface retention of phospho-IGF-IR after prolonged IGF-I stimulation (*Figure 2F,G*), indicating that the IRS-1-induced surface retention of activated IGF-IR is independent on the Tyr phosphorylation of IRS-1.

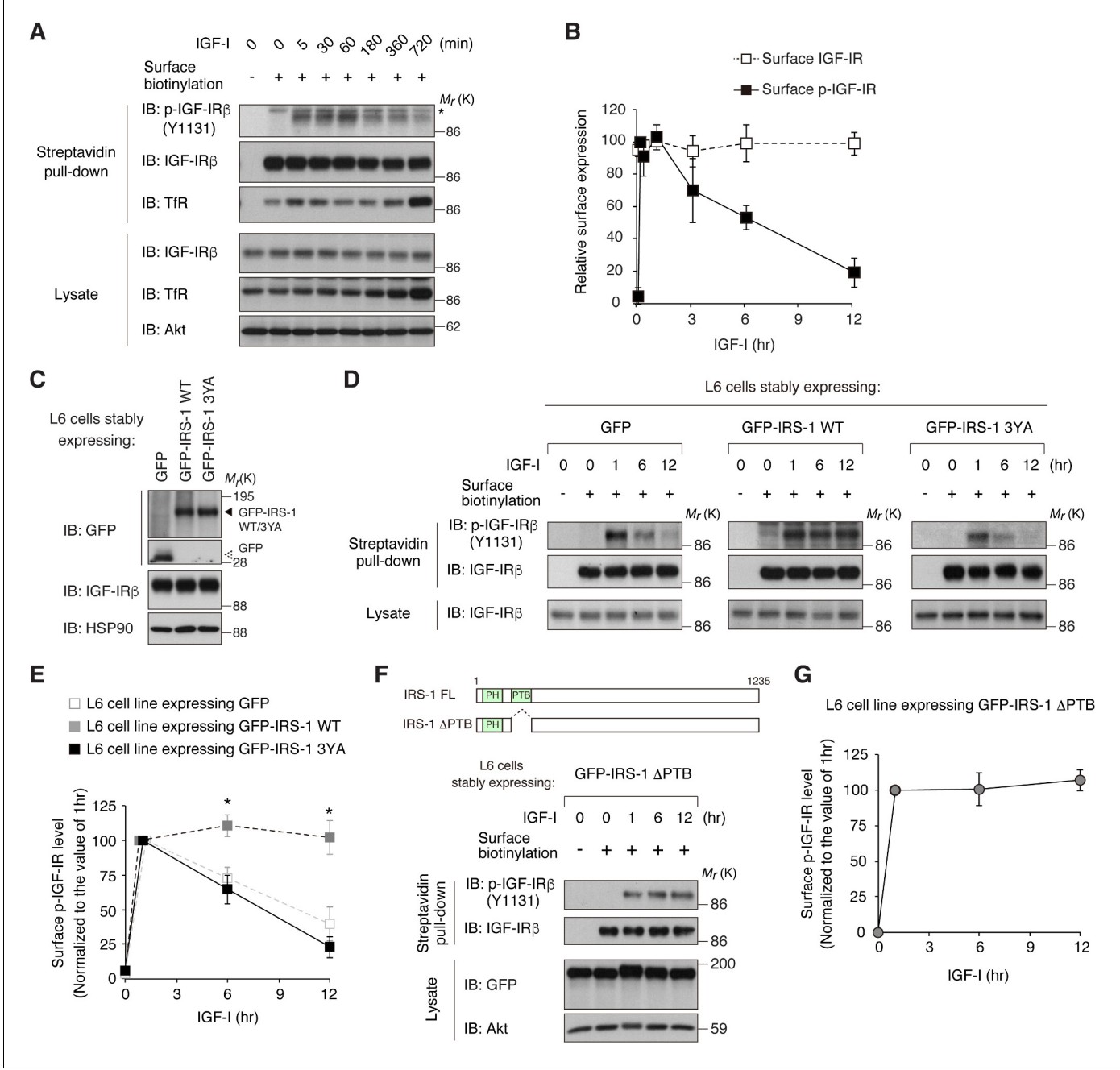

**Figure 2.** IRS-1 promotes cell surface retention of activated IGF-IR via its YxxΦ motifs. (**A**) Changes in cell surface IGF-IR following IGF-I stimulation in L6 cells were analyzed by surface biotinylation assay. Transferrin receptor (TfR) was evaluated as a loading control for cell surface protein. (**B**) Immunoblots of surface IGF-IR for (**A**) were quantified and the graph is shown as mean ±SEM of four independent experiments. (**C**) Immunoblotting of GFP-IRS-1 wild-type (WT) and 3YA mutant in lysates from L6 cells stably expressing GFP, GFP-IRS-1 WT, or GFP-IRS-1 3YA. (**D**) Changes in surface phospho-IGF-IR following IGF-I stimulation were analyzed in L6 cells stably expressing GFP, GFP-IRS-1 WT, or GFP-IRS-1 3YA by surface biotinylation assay. (**E**) Immunoblots of surface IGF-IR for (**D**) were quantified and the graph is shown as mean ±SEM of four independent experiments. Differences were analyzed by ANOVA and the Tukey *post hoc* test. *p<0.05 versus GFP. (**F, G**) Changes in surface phospho-IGF-IR following IGF-I stimulation were analyzed in L6 cells stably expressing GFP-IRS-1 ΔPTB by surface biotinylation assay (**F**). Immunoblots of surface IGF-IR for (**F**) were quantified and the graph is shown as mean ±SEM of three independent experiments (**G**).

DOI: https://doi.org/10.7554/eLife.32893.006

The following figure supplement is available for figure 2:

**Figure supplement 1.** Expression of IRS-1, but not IRS-2, inhibits the down-regulation of activated IGF-IR induced by long-term IGF-I stimulation.

DOI: https://doi.org/10.7554/eLife.32893.007

## Internalization of active IGF-IR is dependent on the clathrin/AP2-mediated endocytic pathway

We investigated whether long-term IGF-I-induced reduction in activated IGF-IR depends on CME. In clathrin-depleted cells, the reduction in phospho-IGF-IR observed after long-term IGF-I stimulation was completely blocked (*Figure 3A*). Similarly, the knockdown of AP2 (μ2), but not of another clathrin adaptor AP1 (μ1), inhibited the reduction of phospho-IGF-IR (*Figure 3B* and *Figure 3—figure supplement 1A*).

The canonical CME model of RTKs involves their rapid depletion from the cell surface in response to the ligands (*Goh and Sorkin, 2013*). Surface biotinylation analysis in *Figure 2A* revealed that the total amount of IGF-IR at the cell surface is not changed by IGF-I. Surface IGF-IR level reflects the balance between endocytosis, recycling, and the transport of newly synthesized receptor to the plasma membrane. When the recycling was inhibited by primaquine (*van Weert et al., 2000*), surface IGF-IR levels were reduced by IGF-I treatment within 1 hr, and phospho-IGF-IR levels followed this time-dependent changes (*Figure 3C*), indicating that IGF-I indeed triggers IGF-IR endocytosis from cell surface and that the recycling contributes to the apparent surface maintenance of IGF-IR. We also assessed the contribution of newly synthesized IGF-IR by using cycloheximide which could inhibit the increase in precursor IGF-IR observed in long-term IGF-I-stimulated cells. IGF-I reduced surface IGF-IR in the presence of cycloheximide (*Figure 3—figure supplement 2A*). These observations support the notion that transport mechanisms other than endocytosis contribute to the maintenance of surface IGF-IR level.

Protein tyrosine phosphatase 1B (PTP1B), an endoplasmic reticulum-resident phosphatase, has been reported to down-regulate IGF-IR by dephosphorylation (*Buckley et al., 2002*). We tested the possible involvement of PTP1B in long-term IGF-I-induced reduction in activated IGF-IR by using the substrate-trapping mutant (PTP1B D181A). Phosphorylation levels of IGF-IR observed 1 hr after IGF-I treatment and the subsequent reduction at the later period (6 hr) were comparable for both PTP1B D181A-expressing and non-expressing cells as revealed by immunofluorescence (*Figure 3—figure supplement 2B*), indicating a negligible role of PTP1B in the down-regulation of phospho-IGF-IR in our observation.

To directly monitor the internalized IGF-IR, we stimulated surface-biotinylated cells with IGF-I and then analyzed the internalized IGF-IR fraction (see Materials and methods). It revealed that internalized IGF-IR was detected within 15 min after surface biotinylation (*Figure 3—figure supplement 3A*). Similar results also came from the immunofluorescence analysis of a double-tagged IGF-IR-transfected cells. The IGF-IR-HA-EGFP construct that we developed contains an extracellular HA-tag and intracellular EGFP and can be utilized to directly monitor the internalization by following uptake of anti-HA antibody added to the media prior to ligand treatment (*Figure 3—figure supplement 3B,C*). Internalized fraction of the double-tagged IGF-IR was detected within 15 to 60 min in both IGF-I-stimulated and non-stimulated conditions (*Figure 3—figure supplement 3D,E*). The internalization of IGF-IR observed in the non-stimulated state was not affected by knockdown of AP2 (*Figure 3—figure supplement 3F*), indicating that the basal endocytosis of IGF-IR is not dependent on AP2. In contrast, phospho-IGF-IR was predominantly localized to the cell surface and did not overlap with internalized IGF-IR (HA-positive) within 1 hr in the ligand-stimulated cells (*Figure 3—figure supplement 3D*). At the later period (6 hr), phospho-IGF-IR was detected in LysoTracker-positive compartments (*Figure 3—figure supplement 1B*, left). More importantly, the phospho-IGF-IR targeting to lysosomes was abolished by knockdown of AP2 (*Figure 3—figure supplement 1B*, right; *Figure 3—figure supplement 1C*), suggesting that ligand-activated IGF-IR undergoes AP2-dependent endocytosis.

Using live cell total internal fluorescence microscopy (TIRF-M), we investigated the detailed onset of IGF-IR internalization. The assembly of AP2 into clathrin-coated structures can be monitored by the expression of AP2 σ2 subunit fused with monomeric red fluorescent protein (mRFP) (*Ehrlich et al., 2004*). IGF-IR-EGFP was uniformly distributed within the plasma membrane, and then gradually colocalized with σ2-mRFP 30 min after IGF-I stimulation (*Figure 3D*, left). We also observed similar results in the fixed cells where phospho-IGF-IR was overlapped with AP2 and clathrin (*Figure 3—figure supplement 4A–C*). In more detail, IGF-IR clustered after IGF-I stimulation, and then accumulated in pre-existing AP2-positive spots (*Figure 3—figure supplement 4D*). EGFP-fused EGFR, which is a representative RTK showing rapid CME, was rapidly re-distributed into AP2-

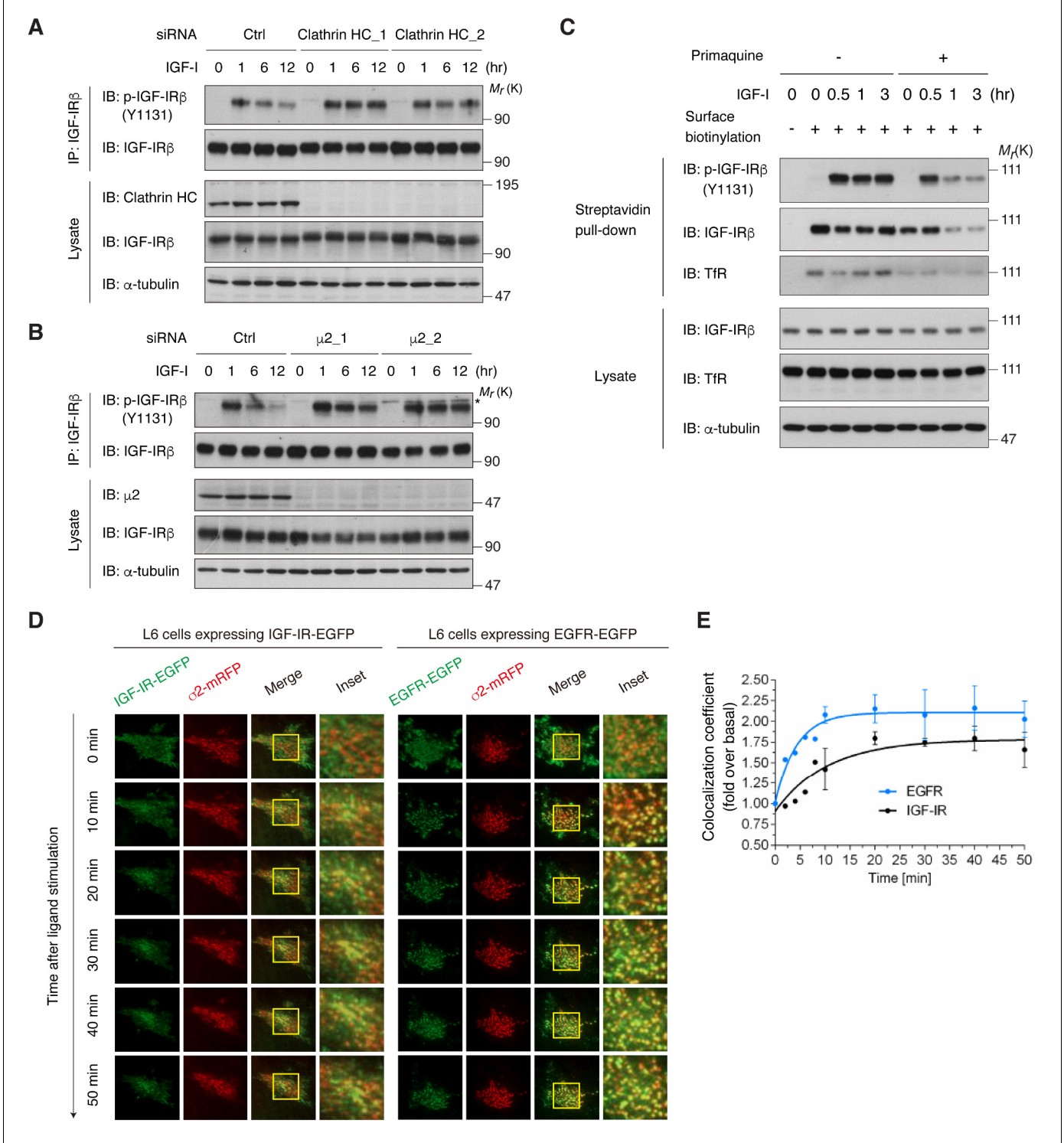

**Figure 3.** Internalization of activated IGF-IR is dependent on the clathrin/AP2-mediated endocytic pathway. (**A**) Knockdown of clathrin heavy chain (HC) by two different siRNAs blocked long-term IGF-I-induced reduction of phospho-IGF-IR in L6 cells. Ctrl, control. The data are representative of three independent experiments. (**B**) Knockdown of the μ2 subunit of AP2 by two different siRNAs blocked long-term IGF-I-induced reduction of phospho-IGF-IR in L6 cells. Asterisk indicates a nonspecific band. The data are representative of at least three independent experiments. The μ2_1 siRNA was used in further experiments. (**C**) Changes in cell surface IGF-IR following IGF-I stimulation in L6 cells that were pre-treated with primaquine were analyzed by surface biotinylation assay. (**D**) Live cell TIRF-M imaging of L6 cells expressing IGF-IR-EGFP (left) or EGFR-EGFP (right) together with σ2-mRFP, which were stimulated for the indicated times with IGF-I or EGF, respectively. A representative region at higher magnification outlined by yellow

*Figure 3 continued on next page*

*Figure 3 continued*

rectangles is also shown in insets. (**E**) Quantification of colocalization between IGF-IR (black line) or EGFR (blue line) and AP2 in (**D**). Mean (fold over the value at 0 min)± SD is shown (n = 7 cells). The data are representative of three independent experiments.

DOI: https://doi.org/10.7554/eLife.32893.008

The following figure supplements are available for figure 3:

**Figure supplement 1.** AP2, but not AP1, is required for the targeting of activated IGF-IR from the plasma membrane into lysosomes.

DOI: https://doi.org/10.7554/eLife.32893.009

**Figure supplement 2.** Effects of cycloheximide treatment and PTP1B D181A expression on surface IGF-IR changes after the ligand exposure.

DOI: https://doi.org/10.7554/eLife.32893.010

**Figure supplement 3.** Chase of internalized IGF-IR.

DOI: https://doi.org/10.7554/eLife.32893.011

**Figure supplement 4.** Colocalization of IGF-IR with AP2 in response to the ligand treatment.

DOI: https://doi.org/10.7554/eLife.32893.012

positive spots after EGF stimulation (*Figure 3C*, right). Intriguingly, quantitative analyses revealed that IGF-I-induced increase in the colocalization rate of IGF-IR with AP2 was significantly slower than EGFR (*Figure 3D*).

## IRS-1 inhibits the AP2-dependent internalization of IGF-IR

Expression of IRS-1 WT, but not 3YA mutant, induced surface retention of activated IGF-IR (*Figure 2D,E*), which phenocopies that of AP2 knockdown (*Figure 3B*). We next asked whether IRS-1 could disrupt IGF-IR internalization. To clearly evaluate ligand-dependent receptor internalization, we performed surface biotinylation assay of IGF-I-stimulated cells when the recycling was inhibited by primaquine. While surface IGF-IR levels were gradually reduced after the ligand stimulation in cells expressing GFP and GFP-IRS-1 3YA, such reduction turned to be slower in cells expressing GFP-IRS-1 WT (*Figure 4A*). In addition, the TIRF-M revealed that expression of GFP-IRS-1 WT, but not of 3YA mutant, significantly inhibited the targeting of phospho-IGF-IR in AP2-positive spots with diffused localization of phospho-IGF-IR (*Figure 4C,D*), indicating that the IRS-1 binding to AP2 inhibits the ligand-induced association of IGF-IR with AP2-positive spots.

Since AP2 regulates CME of various membrane cargoes, we next asked if ectopic expression of IRS-1 affects endocytosis of other cargoes. The internalization of transferrin receptor (TfR), integrin, and EGFR, which are endocytosed through CME, was evaluated. We analyzed the endocytosis of TfR, which has no physical interaction with IGF-IR (*Figure 4—figure supplement 1A*), by measuring uptake of fluorescent-labeled transferrin. Overexpression of IRS-1 did not affect the uptake of transferrin (*Figure 4—figure supplement 1B*). Integrins including β1 are involved in the crosstalk with IGF-IR signaling (*Kiely et al., 2005*). Surface level and internalization of integrin β1 were assessed by labeling cell surface with anti-integrin β1 antibody and chasing its uptake (see Materials and methods). In L6 cells stably expressing integrin β1 which modestly interacts with IGF-IR (*Figure 4—figure supplement 2A*), surface expression of integrin β1 was not statistically different between IRS-1-expressing and control cells (*Figure 4—figure supplement 2B*; p=0.188). The incorporated amount of anti-integrin β1 antibody was partially reduced in IRS-1-expressing cells (*Figure 4—figure supplement 2C,D*). We also examined the endocytosis of EGFR induced by low-dose EGF, which is dependent on CME (*Sigismund et al., 2008*), by observing localization of the transfected EGFR-GFP. Modest delay of EGFR endocytosis was observed at the early period of EGF stimulation in IRS-1-expressing cells (*Figure 4—figure supplement 2E–F*). These observations indicate that IRS-1 can influence endocytosis of receptors other than IGF-IR.

We also confirmed that the number of AP2 spots at TIRF field was not affected by the expression of IRS-1 (*Figure 4—figure supplement 1A*). By using TIRF-M, we noticed that GFP-IRS-1 colocalizing with AP2 is localized to submembraneous actin fibers, which possess critical roles in CME (*Kaksonen et al., 2006*) (*Figure 4—figure supplement 1D*).

If endogenous IRS-1 inhibits IGF-IR internalization, knockdown of IRS-1 would accelerate the process of active IGF-IR reduction triggered by long-term IGF-I stimulation. IRS-1 knockdown in L6 cells resulted in a faster reduction of phospho-IGF-IR (~2 fold) with a partial decrease in IGF-IR level (*Figure 5A,B*; reduction rate of p-IGF-IR from 1 to 3 hr of IGF-I treatment (value ± SEM (/hr)), siCtrl, 7.8 ± 2.2; siIRS1_1, 15.1 ± 1.7; siIRS1_2, 17.2 ± 2.4; p<0.05 versus siCtrl). Furthermore, phospho-IGF-

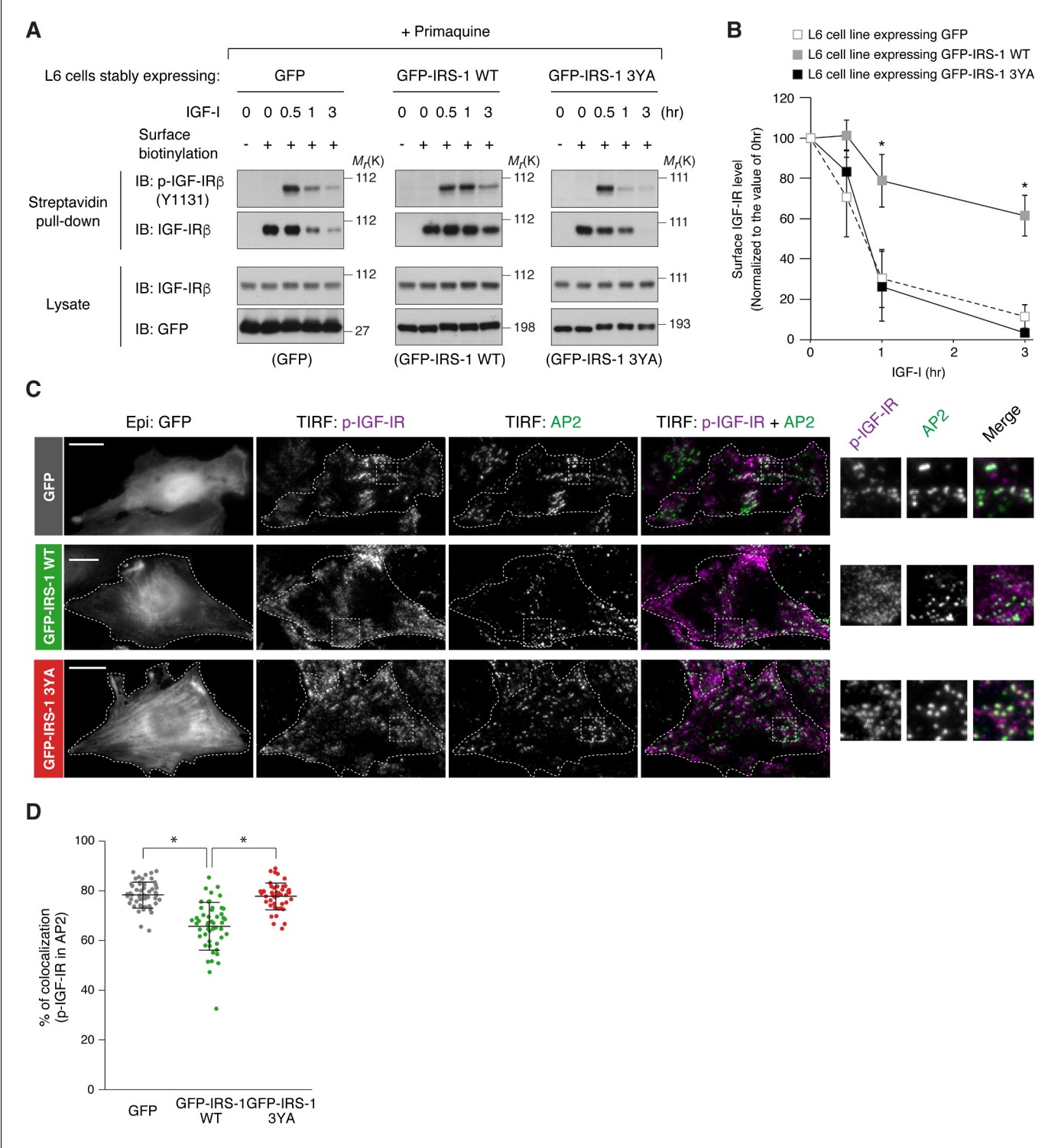

**Figure 4.** IRS-1 inhibits the recruitment of active IGF-IR into clathrin-coated structures. (A) Changes in surface phospho-IGF-IR following IGF-I stimulation in the presence of primaquine were analyzed in L6 cells stably expressing GFP, GFP-IRS-1 WT, or GFP-IRS-1 3YA by surface biotinylation assay. (B) Immunoblots of surface IGF-IR for (A) were quantified and the graph is shown as mean ±SEM of three independent experiments. Differences were analyzed by ANOVA and the Tukey *post hoc* test. *p<0.05 versus GFP. (C) L6 cells stably expressing IGF-IR-FLAG were transfected with the plasmid expressing GFP, GFP-IRS-1 WT, or GFP-IRS-1 3YA. The cells were stimulated with IGF-I for 1 hr. Colocalization of phospho-IGF-IR with AP2 was analyzed in the immunostained cells by TIRF-M. Insets show representative regions at higher magnification. Bar, 10 µm. (D) Quantification of colocalization between phospho-IGF-IR and AP2 in (C). The colocalization rate in each transfected cell is plotted and mean ±SD is shown (n > 50 cells in

*Figure 4 continued on next page*

*Figure 4 continued*

each condition). The data are representative of three independent experiments. Differences were analyzed by ANOVA and the Tukey *post hoc* test.
*p<0.05.

DOI: https://doi.org/10.7554/eLife.32893.013

The following figure supplements are available for figure 4:

**Figure supplement 1.** Effects of IRS-1 overexpression on AP2-positive spot formation and endocytosis of transferrin receptor.

DOI: https://doi.org/10.7554/eLife.32893.014

**Figure supplement 2.** Effects of IRS-1 overexpression on endocytosis of integrin β1 and EGFR.

DOI: https://doi.org/10.7554/eLife.32893.015

IR accumulated in lysosomes in IRS-1-depleted cells 1 hr after IGF-I stimulation when phospho-IGF-IR is predominantly localized to the plasma membrane in control cells (*Figure 5—figure supplement 1A,B*). Notably, the partial reduction of total IGF-IR levels observed in IRS-1-depleted cells was rescued by the combined knockdown of AP2 (*Figure 5C,D*). The accelerated reduction of phospho-IGF-IR after IGF-I stimulation in IRS-1-depleted cells was also attenuated by the combined knockdown of AP2 (*Figure 5E,F*), indicating that knockdown of IRS-1 accelerates IGF-I-induced IGF-IR internalization as well as reducing IGF-IR levels in an AP2-dependent manner. These results further support the notion that IRS-1 inhibits AP2-mediated internalization of IGF-IR and its long-term attenuation.

## mTOR-dependent degradation of IRS-1 is required for the initiation of IGF-IR internalization

Previous studies have demonstrated a negative feedback loop in which long-term IGF/insulin stimulation induces the degradation of IRS-1 in a PI3K/mTOR complex 1 (mTORC1)-sensitive and proteasome-dependent fashion (*Harrington et al., 2004*; *Haruta et al., 2000*). In L6 cells, the amount of IRS-1 was significantly reduced 3 to 6 hr after IGF-I stimulation with a concomitant increase in its phosphorylation (*Figure 6A*). Pharmacological inhibition of mTORC1 with rapamycin or Torin1 blunted the IRS-1 degradation (*Figure 6B*). Simultaneously, the reduction of phospho-IGF-IR after IGF-I stimulation was also blocked by mTORC1 inhibition (*Figure 6B,C*). TIRF-M analysis revealed that phospho-IGF-IR was less clustered, and overlapped very little with AP2 in Torin1-treated cells (*Figure 6D,E*). In IRS-1-depleted cells, phospho-IGF-IR levels were decreased after long-term IGF-I stimulation even in the presence of Torin1 (*Figure 6F,G*). Collectively, these results suggest that the degradation of IRS-1 via mTORC1-mediated feedback loop is required for the internalization of activated IGF-IR.

## IRS-1 is critical for sustained activation of Akt and inactivation of FoxO

Given that CME affects signaling duration, we tested the role of IRS-1 in the temporal changes in downstream pathways of IGF-IR. Like phospho-IGF-IR, IGF-I-induced phosphorylation of Akt was sustained within 1 hr with a gradual decrease afterwards in L6 cells (*Figure 6A*). Ectopic expression of IRS-1 WT, however, significantly prolonged the phosphorylation of Akt in response to IGF-I (*Figure 7A,B*). Phosphorylation of FoxO1, a transcription factor targeted by Akt (*Calnan and Brunet, 2008*), was also prolonged in IRS-1 WT-overexpressing cells. These described effects on Akt and FoxO1 were not observed in cells overexpressing IRS-1 3YA mutant (*Figure 7A,B*). In addition, overexpression of IRS-2 did not prolong the IGF-I-dependent Akt phosphorylation with a slight increase in its maximum response (*Figure 7—figure supplement 1A*).

We next assessed the role of endogenous IRS-1 in the Akt-FoxO signaling duration by using siRNA-mediated knockdown of IRS-1. In IRS-1-depleted cells, the phosphorylation of Akt showed a very transient pattern with the acute decrease in the later period of IGF-I stimulation (*Figure 7C,D*). During the shorter stimulation, IRS-1 depletion had a minimal effect on the Akt phosphorylation, which may be explained by the compensatory increase in IRS-2 protein (*Figure 7—figure supplement 1B*). The phosphorylation of FoxO1 was transient in IRS-1-depleted cells while it was stable (phospho-S256 in FoxO1) or accumulated (phospho-T24 in FoxO1 or T32 in FoxO3) in control cells (*Figure 7C,D*). The shift from sustained to transient phosphorylation of Akt in IRS-1-depleted cells was completely recovered by the rescue expression of IRS-1 (*Figure 7—figure supplement 1C*).

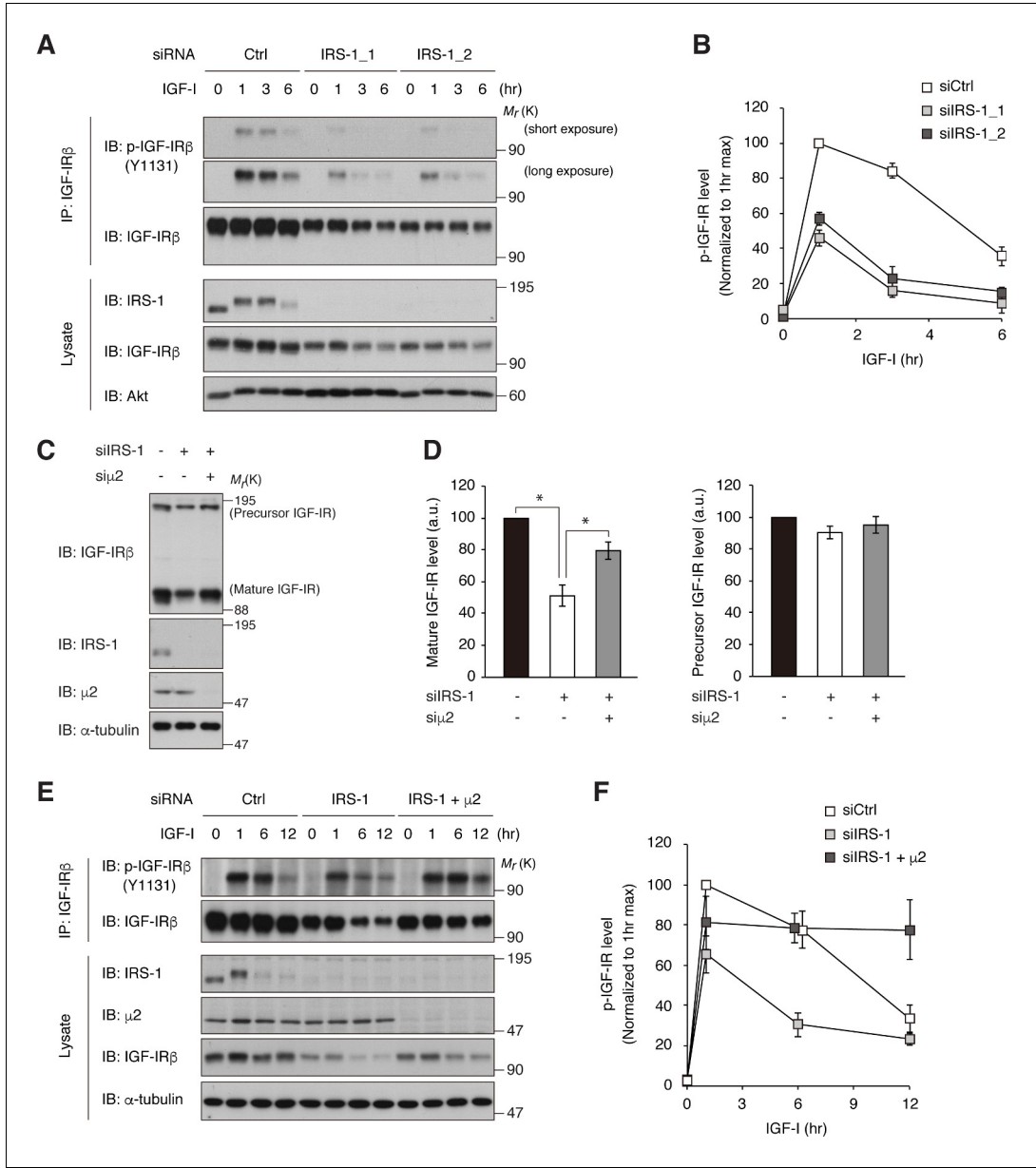

**Figure 5.** Depletion of IRS-1 accelerates AP2-dependent internalization of IGF-IR. (**A, B**) L6 cells transfected with non-targeting (Ctrl) or IRS-1 siRNA were stimulated with IGF-I for the indicated time. Phosphorylation of IGF-IR was analyzed by immunoprecipitation and immunoblotting with the indicated antibodies (**A**). Both short and long exposed immunoblots of phospho-IGF-IR are shown. Immunoblots of phospho-IGF-IR for (**A**) were quantified and the graph is shown as mean ±SEM of four independent experiments (**B**). (**C, D**) L6 cells were transfected with IRS-1 siRNA combined with or without μ2 siRNA. The indicated proteins were analyzed by immunoblotting (**C**). Immunoblots of mature and precursor IGF-IR for (**C**) were quantified and the graph is shown as mean ±SEM of four independent experiments (**D**). Differences were analyzed by ANOVA and the Tukey *post hoc* test. *p<0.05. a. u., arbitrary unit. (**E, F**) L6 cells were transfected with non-targeting or IRS-1 siRNA combined with or without μ2 siRNA. The cells were stimulated with IGF-I for the indicated time. Phosphorylation of IGF-IR was analyzed by immunoprecipitation and immunoblotting with the indicated antibodies (**E**). Immunoblots of phospho-IGF-IR for (**E**) were quantified and the graph is shown as mean ±SEM of three independent experiments (**F**).

DOI: https://doi.org/10.7554/eLife.32893.016

The following figure supplement is available for figure 5:

**Figure supplement 1.** IRS-1 inhibits the targeting of IGF-IR into lysosomes.

DOI: https://doi.org/10.7554/eLife.32893.017

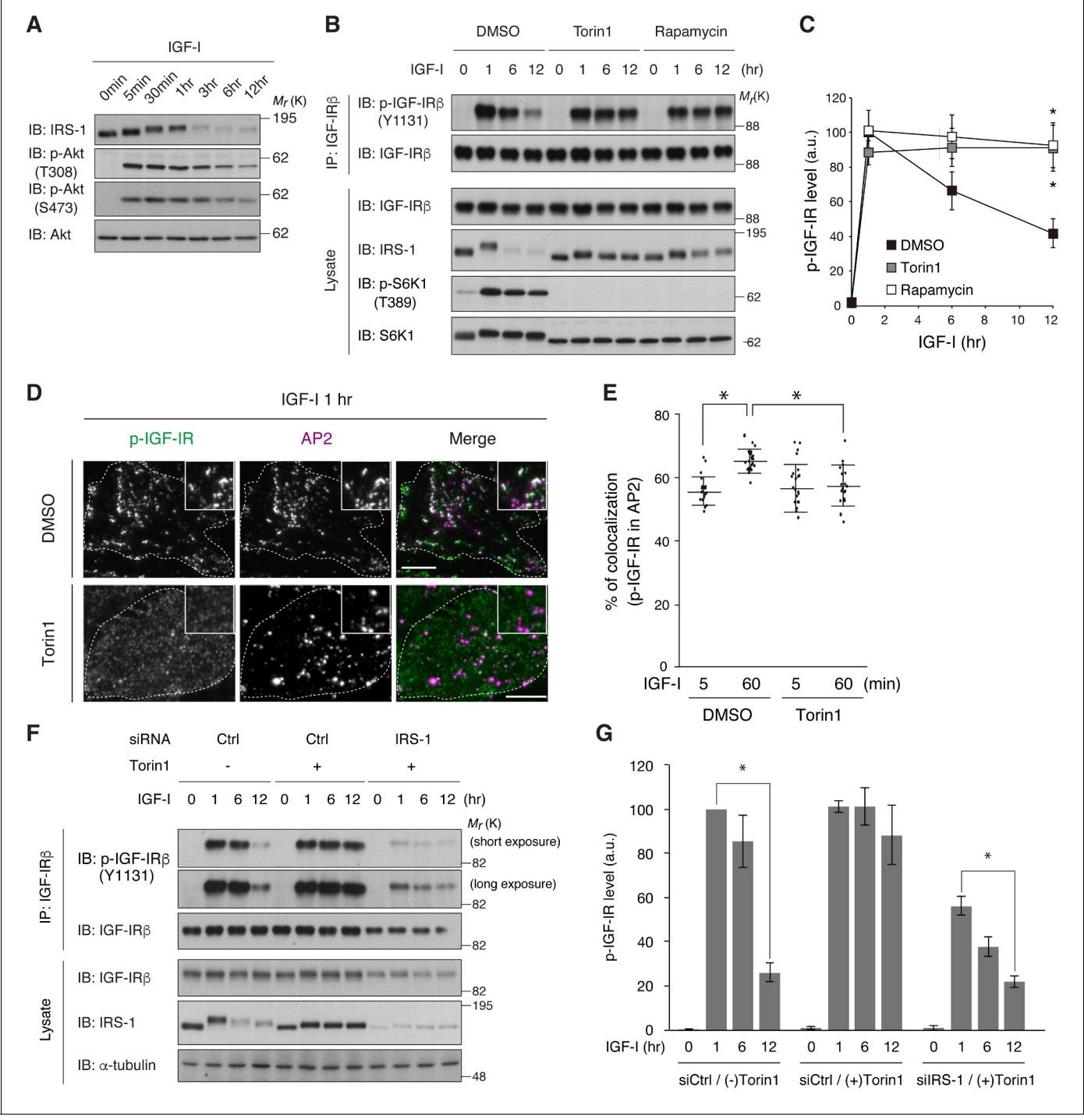

**Figure 6.** mTOR-dependent degradation of IRS-1 is required for the initiation of IGF-IR internalization. (**A**) Changes in IRS-1 and Akt phosphorylation following IGF-I stimulation were analyzed in L6 cells by immunoblotting. (**B, C**) L6 cells were treated with Torin1 or rapamycin followed by IGF-I stimulation. Phosphorylation of IGF-IR was analyzed by immunoprecipitation and immunoblotting with the indicated antibodies (**B**). Immunoblots of phospho-IGF-IR for (**B**) were quantified and the graph is shown as mean ±SEM of four independent experiments (**C**). Differences were analyzed by ANOVA and the Tukey *post hoc* test. *p<0.05. (**D, E**) L6 cells stably expressing IGF-IR-EGFP were treated with or without Torin1 followed by IGF-I stimulation for 1 hr. Colocalization of phospho-IGF-IR with AP2 was analyzed in the immunostained cells by TIRF-M (**D**). Insets show a representative region at higher magnification. Bar, 10 μm. Quantification of colocalization between phospho-IGF-IR and AP2 in (**D**) in each cell is plotted and the means are shown (**E**; n > 25 cells). Differences were analyzed by ANOVA and the Tukey *post hoc* test. *p<0.05. The data are representative of three independent experiments. (**F, G**) L6 cells were transfected with non-targeting or IRS-1 siRNA. The cells were treated with or without Torin1 followed by IGF-I stimulation for the indicated time. Phosphorylation of IGF-IR was analyzed by immunoprecipitation and immunoblotting with the indicated

*Figure 6 continued on next page*

*Figure 6 continued*

antibodies (**F**). Immunoblots of phospho-IGF-IR for (**F**) were quantified and the graph is shown as mean ±SEM of four independent experiments (**G**). Differences were analyzed by ANOVA and the Tukey *post hoc* test. *p<0.05.

DOI: https://doi.org/10.7554/eLife.32893.018

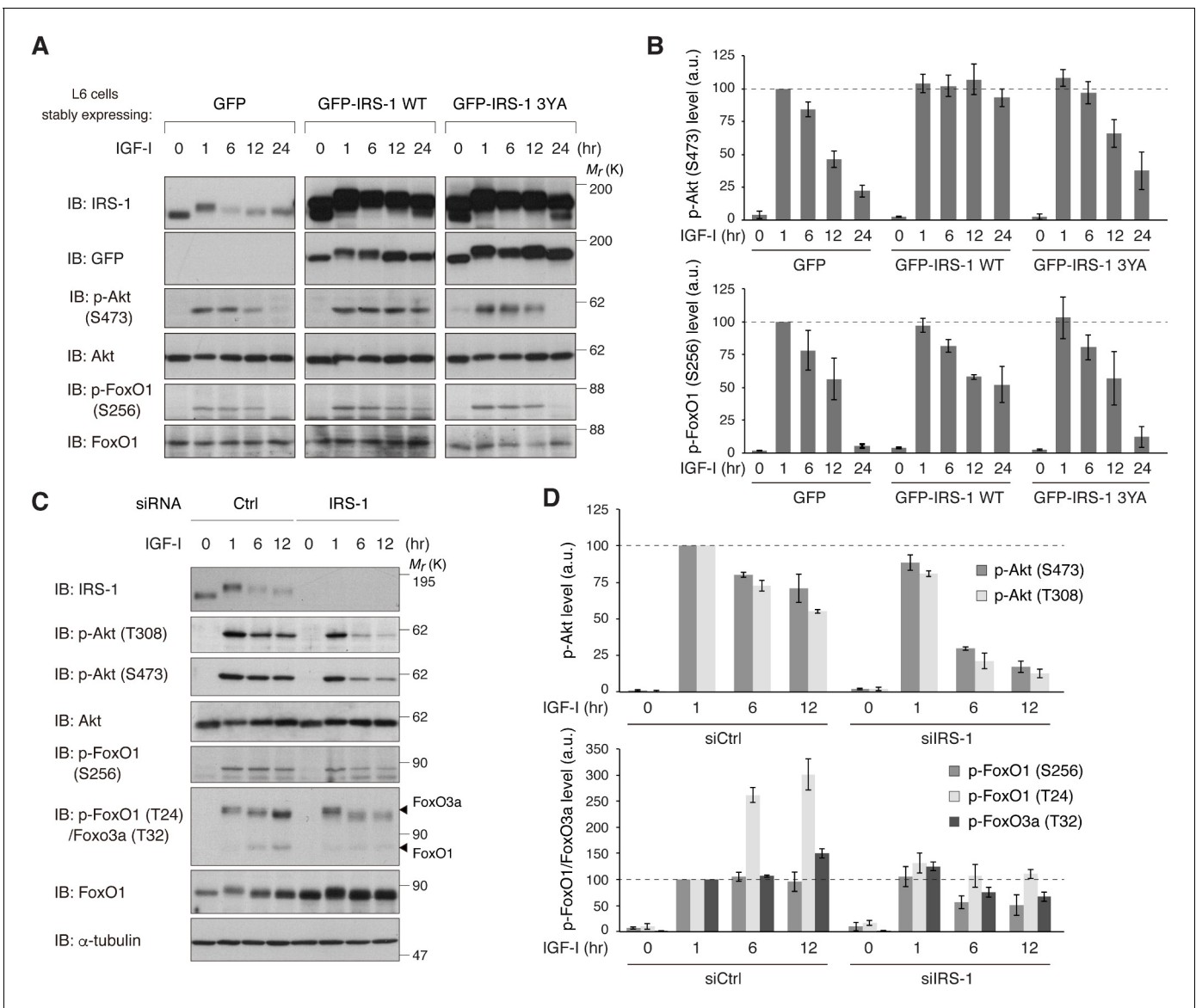

**Figure 7.** IRS-1 is required for sustained activation of Akt and FoxO inactivation in response to IGF-I. (**A, B**) Immunoblotting after treating with IGF-I for the indicated time in L6 cells stably expressing GFP, GFP-IRS-1 WT, or GFP-IRS-1 3YA (**A**). Immunoblots of phospho-Akt (S473) and phospho-FoxO1 (S256) for (**A**) were quantified and the graph is shown as mean ±SEM of three independent experiments (**B**). (**C, D**) Immunoblotting after treating with IGF-I for the indicated time in L6 cells transfected with non-targeting or IRS-1 siRNA (**C**). Immunoblots of phospho-Akt (T308 and S473) and phospho-FoxO (S256 and T24 in FoxO1, and T32 in FoxO3a) for (**C**) were quantified and the graph is shown as mean ±SEM of three independent experiments (**D**).

DOI: https://doi.org/10.7554/eLife.32893.019

The following figure supplement is available for figure 7:

**Figure supplement 1.** Neither overexpression of IRS-2 nor solely blocking of IGF-IR internalization leads to sustained activation of Akt.

DOI: https://doi.org/10.7554/eLife.32893.020

These results indicate a role of IRS-1 in sustaining the Akt-FoxO signaling as well as prolonged surface retention of active IGF-IR.

## IRS-1 is required for efficient down-regulation of FoxO-targeting genes mediated by IGF

Since Akt inhibits the transcriptional activity of FoxOs via their phosphorylation (*Calnan and Brunet, 2008*), we reasoned that sustained activation of Akt in response to IGF could efficiently suppress FoxO-targeting gene expression. Here, we measured the mRNA expression levels of a series of FoxO-regulated genes related to muscle atrophy in which ubiquitin-proteasomal and autophagic protein degradation is enhanced (*Milan et al., 2015*; *Mammucari et al., 2007*; *Moses et al., 2014*; *Stitt et al., 2004*; *Zhao et al., 2007*). In L6 myotubes long-term IGF-I stimulation significantly reduced the mRNA expression level of the two muscle-specific E3 ubiquitin ligases (*Atrogin1 and Murf1*) and recently reported E3 ligases (*Smart* and *Musa1*) as well as autophagy-related genes (*Lc3b* and *Gabarapl1*) (*Figure 8A* and *Figure 8—figure supplement 1A*). These genes were also down-regulated by IGF-I in L6 myoblasts (*Figure 8—figure supplement 1B*). To reveal the contribution of IRS-1 to their expression, we analyzed their mRNA levels in IRS-1-depleted L6 myoblasts. In these cells, IGF-I-induced decrease in the atrophy-related genes was markedly attenuated (*Figure 8B*). We also tested whether IRS-1 knockdown would affect the myotube morphology (*Figure 8—figure supplement 1C*). We confirmed that lentiviral IRS-1 knockdown did not affect the fusion rate (the number of nuclei in myotube fiber) (*Figure 8—figure supplement 1D*). IRS-1-depleted myotubes showed a significant reduction in their diameter (*Figure 8C,D*). These data indicate that IRS-1 depletion leads to insufficient suppression of the FoxO-targeting genes in response to IGF even when Akt is being activated, but in a transient fashion.

## Discussion

The canonical function of IRS proteins is to mediate signaling of IGF-IR to the PI3K-Akt pathway through Tyr phosphorylation (*Figure 9A*) (*White, 2002*). The present results reveal a new role of IRS-1 independent of its Tyr phosphorylation: IRS-1 regulates IGF-IR internalization to produce sustained activation of IGF signaling (*Figure 9B*). IRS-1 binds with AP2 to prevent IGF-IR recruitment into clathrin-coated structures and thus enhance surface retention of activated IGF-IR. This function of IRS-1 in prolonging IGF-IR activity is critical for sustained activation of the PI3K-Akt pathway, and provides a key mechanism for how IGF-IR signaling induces specific biological actions of IGF (*Sacheck et al., 2004*; *Ness and Wood, 2002*; *Bailey et al., 2006*; *Stewart and Rotwein, 1996*). Thus, IRS-1 plays a dual role as a signaling adaptor of IGF-IR and an endocytic regulator of IGF-IR.

The first key finding of the present study is that IRS-1 interacts with AP2 thereby regulating the rate of ligand-dependent internalization of IGF-IR. AP2-mediated recognition of YxxΦ motif in cargos is a critical step for CCP formation (*Traub and Bonifacino, 2013*; *Kadlecova et al., 2017*). Our results indicate that IRS-1 inhibits the recruitment of IGF-IR to CCPs through YxxΦ motifs in IRS-1. We have previously reported that another clathrin adaptor complex AP1 also binds to the same sites of IRS-1 as AP2 (*Yoneyama et al., 2013*). Since AP1 depletion did not prevent the down-regulation of activated IGF-IR (*Figure 3—figure supplement 1A*), the inhibitory effect of IRS-1 on IGF-IR internalization is based on its interaction with AP2. In addition, IRS-1-depleted cells show the fast onset of IGF-IR internalization in response to IGF-I and the partial decrease in IGF-IR levels, both of which are presumably caused by the promotion of AP2-dependent IGF-IR internalization and subsequent degradation (*Figure 5*). These results suggest that IRS-1 is an inhibitory upstream regulator for AP2-dependent internalization of IGF-IR (*Figure 9B*). EGFR and some G-protein coupled receptor/β-arrestin complexes are known to be recruited into pre-existing CCPs after the ligand stimulation (*Rappoport and Simon, 2009*; *Scott et al., 2002*). We observed the similar behavior of IGF-IR in live-cell TIRF-M (*Figure 3—figure supplement 4D*). Notably, less IGF-IR was recruited to AP2-positive spots in the cells ectopically expressing IRS-1 WT, but not 3YA mutant, suggesting that IRS-1 interferes with the recruitment step of IGF-IR to clathrin-coated structures through competing out AP2 from IGF-IR. This will need to be tested by more detailed observation at higher resolution.

The second key finding of this study is that the ability of IRS-1 to promote surface retention of IGF-IR can be separable from Tyr phosphorylation-mediated signaling function of IRS-1. The Tyr residues of the YxxΦ motifs of IRS-1 (Tyr608, 628, and 658) critical for the binding to AP2 are part of

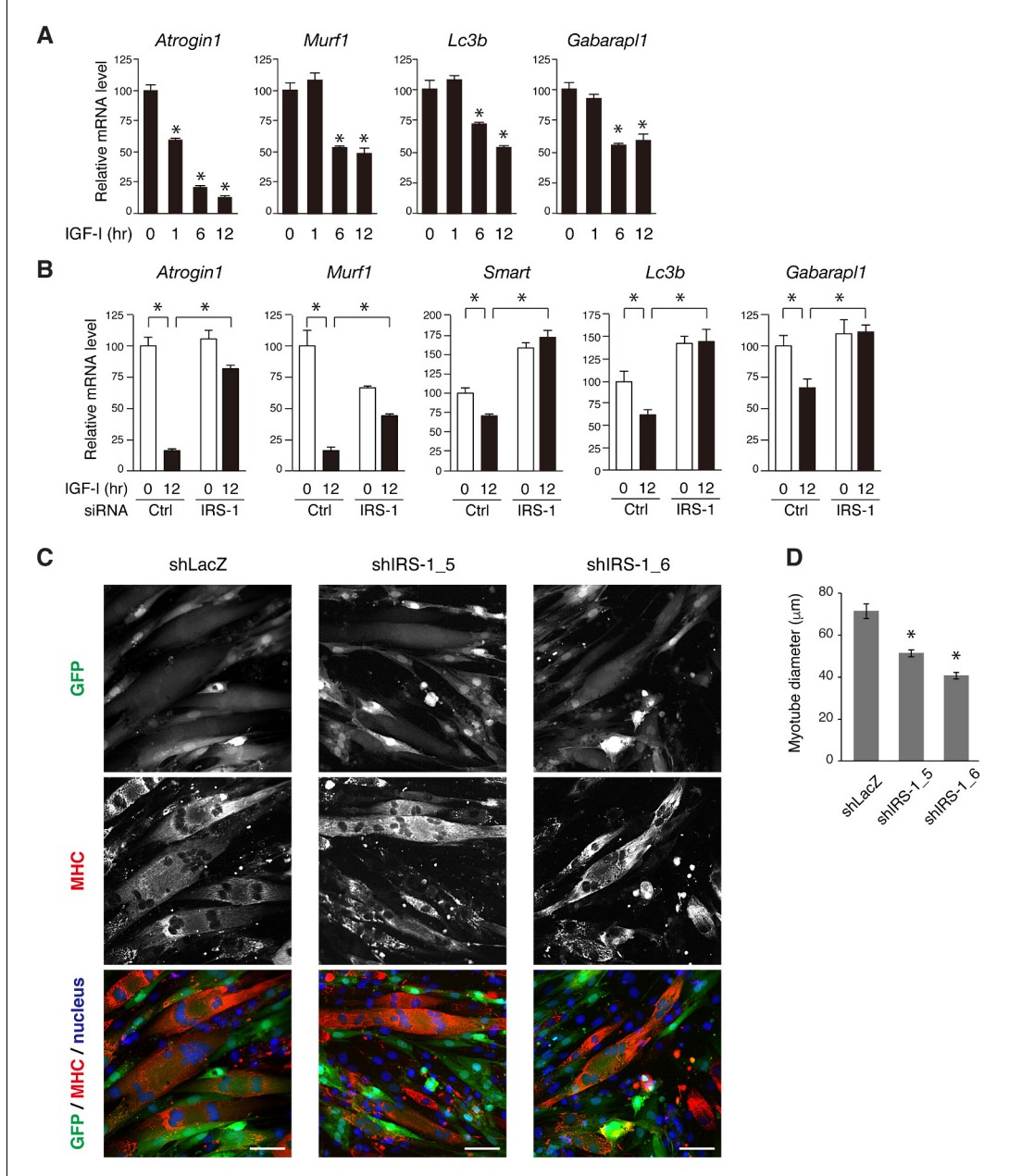

**Figure 8.** IRS-1 is required for efficient down-regulation of atrophy-related genes mediated by IGF-I. (**A**) Quantitative RT-PCR analysis of atrophy-related genes from L6 myotubes stimulated with IGF-I. Data are expressed as fold of the value at 0 hr of IGF-I stimulation. Values are mean ±SEM (n = 3). Differences were analyzed by ANOVA and the Tukey *post hoc* test. *p<0.05 versus IGF-I 0 hr. (**B**) Quantitative RT-PCR analysis of atrophy-related genes from L6 myoblasts transfected with non-targeting or IRS-1 siRNA followed by IGF-I stimulation for 0 or 12 hr. Data are expressed as fold of the value at 0 hr of IGF-I stimulation in cells transfected with control siRNA. Values are mean ±SEM (n = 3). Differences were analyzed by ANOVA and the Tukey *post hoc* test. *p<0.05 versus IGF-I 0 hr. (**C**) L6 myotubes were infected with lentivirus containing LacZ- or IRS-1-targeting shRNA. The infected cells were visualized by GFP expression (green). The fixed cells were immunostained with anti-MHC antibody (red) together with Hoechst nuclear staining (blue). MHC, myosin heavy chain. Bar, 50 μm. (**D**) Measurement of myotube diameter after lentivirus-mediated knockdown of IRS-1 for (**C**). The data are presented as mean ±SEM (n > 100 cells per condition). Differences were analyzed by ANOVA and the Tukey *post hoc* test. *p<0.05.
DOI: https://doi.org/10.7554/eLife.32893.021

The following figure supplement is available for figure 8:

**Figure supplement 1.** Long-term IGF-I stimulation suppresses the FoxO-regulated genes.
DOI: https://doi.org/10.7554/eLife.32893.022

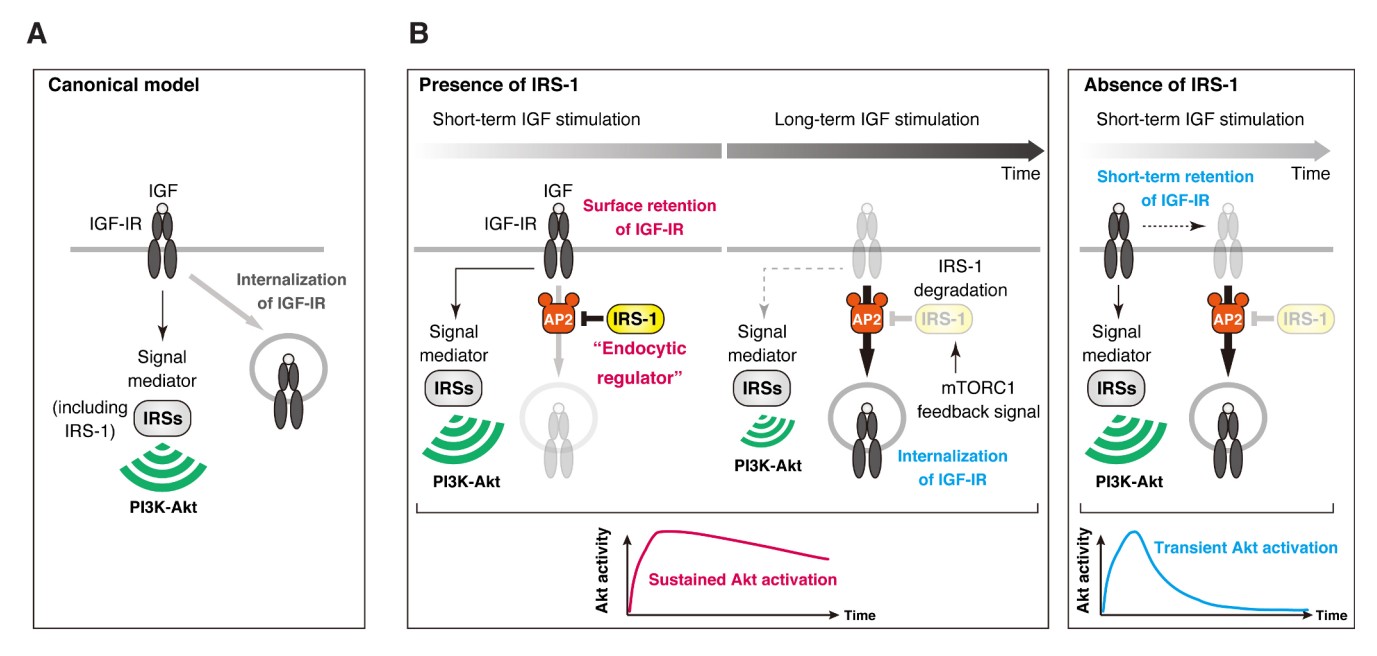

**Figure 9.** Model of IRS-1-mediated control for delayed IGF-IR internalization and its role in the sustained IGF signaling. (**A**) The canonical view in which IRS-1 functions as a signaling mediator of IGF-IR to the PI3K-Akt pathway through their Tyr phosphorylation. The molecular basis for closed interactions between IGF-IR endocytosis and its signaling components has been poorly understood. (**B**) A proposed model for IRS-1-mediated surface retention of IGF-IR and sustained IGF signaling. The ability of IRS-1 to interact with AP2 prolongs the surface retention of active IGF-IR, which is caused by the inhibition of AP2-dependent IGF-IR internalization. After long-term stimulation of IGF, IRS-1 is degraded by mTORC1 feedback signal, which functions as a brake release to trigger the initiation of IGF-IR internalization. Accelerating IGF-IR internalization caused by IRS-1 depletion leads to the shift from sustained to transient Akt signaling.

DOI: https://doi.org/10.7554/eLife.32893.023

phosphorylation sites among multiple Tyr residues in the C-terminus of IRS-1 that mediate the interaction of IRS-1 with PI3K and subsequent activation of PI3K (*Myers et al., 1996*; *Sun et al., 1993*). We showed that ectopic expression of the IRS-1 mutant ΔPTB led to the accumulation of active IGF-IR at cell surface to the same degree as that of IRS-1 WT (*Figure 2*), indicating that IRS-1 inhibits the internalization of IGF-IR in a manner independent of its Tyr phosphorylation. In addition, AP2 would preferentially bind non-phosphorylated IRS-1 since AP2 cannot recognize phosphorylated YxxΦ sequence due to its limited capacity (*Kittler et al., 2008*; *Owen and Evans, 1998*). In line with this, our biochemical analyses support the notion that non-phosphorylated IRS-1 acts as an inhibitory factor for IGF-IR internalization via its interaction with AP2 (*Figure 9B*).

Our observation indicates that ectopic expression of IRS-1 affects endocytosis of receptors other than IGF-IR. As long as we tested, endocytosis of integrin β1 and EGFR, which could interact with IGF-IR, but not of TfR, was inhibited by IRS-1, raising the possibility that IRS-1 influences endocytosis of cargoes in the close proximity of IGF-IR. As observed in our TIRF-M observation (*Figure 4—figure supplement 1D*), a fraction of IRS-1 has been demonstrated to localize to membrane-associated cytoskeleton (*Clark et al., 1998*). IRS-1 may locally regulate the specific cargo recruitment to CCPs through association with a portion of AP2 at the actin cytoskeleton. Indeed, preferred sites of endocytosis have been observed in some cargo proteins (*Grossier et al., 2014*; *Weng et al., 2014*), although the molecular mechanisms of such spatial regulation for IGF-IR and other cargos remain unknown.

In addition to the role of IRS-1 in controlling the rate of IGF-IR internalization, we found that this ability of IRS-1 is negatively regulated by mTORC1 (*Figure 6*). mTORC1 has been reported to suppress IGF-IR activity via its direct substrate Grb10 (*Yu et al., 2011*; *Hsu et al., 2011*). Our findings propose another mode of IGF-IR regulation by mTORC1: mTORC1 feedback signaling leads to the degradation of IRS-1, which functions as a brake release to trigger IGF-IR internalization (*Figure 9B*).

Hence, the time length needed for IRS-1 degradation, which is critically regulated by mTORC1, should determine the initiation timing of IGF-IR internalization.

Receptor endocytosis is now considered to play both negative and positive roles in the downstream signaling (*Goh and Sorkin, 2013*). Our data demonstrated that CME is required for long-term attenuation of activated IGF-IR (*Figure 3*). Previous studies have demonstrated that ligand-activated IGF-IR is ubiquitinated and subsequently undergoes CME for its down-regulation (*Monami et al., 2008*; *Zheng et al., 2012*). In addition, the recycling of IGF-IR has been shown to in part contribute to sustained activation of Akt in response to IGF-I (*Romanelli et al., 2007*). In this study we showed that stable expression of IRS-1 inhibits ligand-dependent internalization of IGF-IR, leading to sustained activation of IGF-IR kinase and the downstream Akt signaling. This effect of IRS-1 on prolonging the Akt signaling is likely based on two independent functions of IRS-1. First, the interaction of IRS-1 with AP2 is required since expression of the IRS-1 mutant 3YA could prolong neither IGF-IR phosphorylation nor Akt phosphorylation in IGF-I-stimulated cells. Second, the ability of IRS-1 to engage PI3K is also necessary because expression of the IRS-1 mutant ΔPTB could prolong phosphorylation of IGF-IR but failed to sustain Akt phosphorylation (*Figure 7—figure supplement 1F*). Similar signaling events were also observed in AP2-depleted cells where IRS-1 degradation, a consequence of negative feedback, was normally induced by long-term IGF-I stimulation (*Figure 7— figure supplement 1D,E*). Notably, the ability to interact with AP2, enhance the surface retention of IGF-IR, and prolong the Akt signaling is specific for IRS-1, but not for IRS-2. Thus, IRS-1 can act as a pivotal modulator for IGF signaling duration via its control of IGF-IR internalization while the downstream signaling activation can be mediated by either IRS-1 or IRS-2 (*Figure 9B*).

It is generally recognized that IGF-IR preferentially mediates growth whereas insulin receptor (IR) functions in glucose homeostasis in spite of the fact that both receptors share common signaling pathways mediated by the IRS proteins (*Accili et al., 1996*; *Liu et al., 1993*; *Nakae et al., 2001*). However, these functional differences between IR and IGF-IR cannot be attributed to characteristics of the receptors themselves, such as their kinetics of ligand binding or their tissue/cellular distribution (*Siddle, 2012*). Moreover, insulin levels fluctuate in response to the nutrients while IGF levels are constantly maintained by circulating IGF binding proteins and by paracrine/autocrine production (*Jones and Clemmons, 1995*). Yet, despite these differences in temporal pattern, this is unlikely to explain the specificity of IGF-IR and IR because even in cell culture these receptors mediate different bioactivities as well as gene expression profiles (*Lammers et al., 1989*; *Palsgaard et al., 2009*), including in a recent study using reconstituted model cell lines solely expressing either receptor (*Cai et al., 2017*). While differential substrate preference for each receptors has been proposed to explain this specificity (*Cai et al., 2017*), both receptors still induce signaling through the PI3K-Akt cascade and involve many IRS proteins (*White, 2002*; *Taniguchi et al., 2006*). In addition, the Akt signaling cascade itself can produce different temporal dynamics in response to specific stimuli and to induce different cellular outcomes (*Gross and Rotwein, 2016*; *Kubota et al., 2012*). Our study demonstrates that the IGF-IR pathway encodes prolonged Akt activation via IRS-1-mediated delay of IGF-IR internalization (*Figure 9B*). In contrast, IR has been shown to undergo rapid CME in response to insulin (*Choi et al., 2016*; *Morcavallo et al., 2012*). These observations raise the possibility that the bioactive difference between IGF-IR and IR arises in part through their differential temporal activation of the PI3K-Akt pathway governed by CME kinetics unique to each receptor. In this context, future studies could productively address whether and how the CME of IGF-IR and IR are selectively regulated, which is also a general issue in the context of CME selectivity for multiple cargos (*Grossier et al., 2014*; *Weng et al., 2014*). Notably, *Choi et al. (2016)* revealed that IR, but not IGF-IR, uses the receptor-associated adaptor BUBR1/MAD2 to facilitate rapid CME by recruiting AP2 to IR. We are likely to better understand the role of differential endocytic regulation of IGF-IR and IR in temporal dynamics of the PI3K-Akt pathway when we identify the specific adaptors for IGF-IR and IR that engage their CME, and determine their relationship with IRS-1.

Our results demonstrate that the prolonged Akt signaling elicited by IRS-1-mediated surface retention of IGF-IR affects the FoxO-targeting gene expression. Long-term action of IGF is fundamental for various physiological aspects including growth control and neural cell survival (*Ness and Wood, 2002*; *Gross and Rotwein, 2016*; *Stewart and Rotwein, 1996*). Thus, IRS-1-mediated delay of IGF-IR internalization is likely to be a common mechanism for long-term IGF actions.

# Materials and methods

**Key resources table**

| Reagent type (species) or resource | Designation | Source or reference | Identifiers | Additional information |
|---|---|---|---|---|
| Strain, strain background (*Escherichia coli*) | BL21 | Agilent Technologies | Agilent Technologies: 200133 | |
| Strain, strain background (*Escherichia coli*) | BL21-CodonPlus(DE3)-RIL | Agilent Technologies | Agilent Technologies: 230245 | |
| Cell line (*Rattus norvegicus*) | L6 | ATCC | ATCC: CRL-1458; RRID: CVCL_0385 | |
| Cell line (*Homo sapiens*) | 293T | ATCC | ATCC: CRL-3216; RRID: CVCL_0063 | |
| Cell line (*Homo sapiens*) | PLAT-E | PMID: 10871756 | RRID: CVCL_B488 | A kind gift from T. Kitamura, The University of Tokyo |
| Antibody | Rabbit polyclonal anti-phospho-IGF-IRβ (Tyr1131) | Cell Signaling Technology | Cell Signaling Technology: 3021; RRID: AB_331578 | IB 1:1000; IF 1:200 |
| Antibody | Rabbit monoclonal anti-phospho-IGF-IRβ (Tyr980) | Cell Signaling Technology | Cell Signaling Technology: 4568; RRID: AB_2122279 | IB 1:1000 |
| Antibody | Rabbit polyclonal anti-phospho-IGF-IRβ (Tyr1316) | Cell Signaling Technology | Cell Signaling Technology: 6113; RRID: AB_10545762 | IB 1:1000 |
| Antibody | Rabbit monoclonal anti-IGF-IRβ | Cell Signaling Technology | Cell Signaling Technology: 9750; RRID: AB_10950969 | IF 1:200 |
| Antibody | Rabbit polyclonal anti-Akt | Cell Signaling Technology | Cell Signaling Technology: 9272; RRID: AB_329827 | IB 1:1000 |
| Antibody | Rabbit polyclonal anti-phospho-Akt (Thr308) | Cell Signaling Technology | Cell Signaling Technology: 9275; RRID: AB_329828 | IB 1:1000 |
| Antibody | Rabbit polyclonal anti-phospho-Akt (Ser473) | Cell Signaling Technology | Cell Signaling Technology: 9271; RRID: AB_329825 | IB 1:1000 |
| Antibody | Rabbit monoclonal anti-phospho-p70 S6K (Thr389) | Cell Signaling Technology | Cell Signaling Technology: 9234; RRID: AB_2269803 | IB 1:1000 |
| Antibody | Rabbit polyclonal anti-phospho-FoxO1 (Thr24) /FoxO3a (Thr32) | Cell Signaling Technology | Cell Signaling Technology: 9464; RRID: AB_329842 | IB 1:1000 |
| Antibody | Rabbit polyclonal anti-phospho-FoxO1 (Sere256) | Cell Signaling Technology | Cell Signaling Technology: 9461; RRID: AB_329831 | IB 1:1000 |
| Antibody | Rabbit monoclonal anti-FoxO1 | Cell Signaling Technology | Cell Signaling Technology: 2880; RRID: AB_2106495 | IB 1:1000 |
| Antibody | Rabbit polyclonal anti-IGF-IRα | Santa Cruz Biotechnology | Santa Cruz Biotechnology: sc-712; RRID: AB_671788 | IB 1:1000 |
| Antibody | Rabbit polyclonal anti-IGF-IRβ | Santa Cruz Biotechnology | Santa Cruz Biotechnology: sc-713; RRID: AB_671792 | IB 1:1000; IP 1:200 |
| Antibody | Rabbit polyclonal anti-IRS-2 | Santa Cruz Biotechnology | Santa Cruz Biotechnology: sc-8299; RRID: AB_2125783 | IB 1:1000 |
| Antibody | Mouse monoclonal anti-clathrin HC | Santa Cruz Biotechnology | Santa Cruz Biotechnology: sc-12734; RRID: AB_627263 | IB 1:1000 |
| Antibody | Mouse monoclonal anti-α-adaptin | Santa Cruz Biotechnology | Santa Cruz Biotechnology: sc-17771; RRID: AB_2274034 | IB 1:1000; IF 1:200 |
| Antibody | Rabbit polyclonal anti-p70 S6K | Santa Cruz Biotechnology | Santa Cruz Biotechnology: sc-230; RRID: AB_632156 | IB 1:1000 |

*Continued on next page*

*Continued*

| Reagent type (species) or resource | Designation | Source or reference | Identifiers | Additional information |
|---|---|---|---|---|
| Antibody | Mouse monoclonal anti-HSP90 | Santa Cruz Biotechnology | Santa Cruz Biotechnology: sc-7947; RRID: AB_2121235 | IB 1:2000 |
| Antibody | Rabbit polyclonal anti-γ-adaptin | Santa Cruz Biotechnology | Santa Cruz Biotechnology: sc-10763; RRID: AB_2058329 | IB 1:1000 |
| Antibody | Mouse monoclonal anti-GFP | Santa Cruz Biotechnology | Santa Cruz Biotechnology: sc-9996; RRID: AB_627695 | IB 1:1000; IP 1:200 |
| Antibody | Mouse monoclonal anti-ubiquitin (P4D1) | Santa Cruz Biotechnology | Santa Cruz Biotechnology: sc-8017; RRID: AB_628423 | IB 1:200 |
| Antibody | Mouse monoclonal anti-FLAG M2 | Sigma-Aldrich | Sigma-Aldrich: F3165; RRID: AB_259529 | IB 1:2000 |
| Antibody | Anti-FLAG M2 agarose affinity gel | Sigma-Aldrich | Sigma-Aldrich: A2220; RRID: AB_10063035 | |
| Antibody | Mouse monoclonal anti-α-tubulin (DM1A) | Sigma-Aldrich | Sigma-Aldrich: T6199; RRID: AB_477583 | IB 1:2000 |
| Antibody | Mouse monoclonal anti-phospho-Tyr (4G10) | Sigma-Aldrich | Sigma-Aldrich: 05-1050X; RRID: AB_916370 | IB 1:1000 |
| Antibody | Rabbit polyclonal anti-IRS-1 | Upstate | Upstate: 06-248; RRID: AB_2127890 | IB 1:1000 |
| Antibody | Mouse monoclonal anti-myosin heavy chain | Upstate | Upstate: 05-716; RRID: AB_309930 | IF 1:200 |
| Antibody | Mouse monoclonal anti-Myc | Upstate | Upstate: 05-419; RRID: AB_309725 | IF 1:200 |
| Antibody | Rabbit polyclonal anti-p85 PI3 kinase | Upstate | Upstate: 06-195; RRID: AB_310069 | IB 1:1000 |
| Antibody | Mouse monoclonal anti-μ2 | BD Transduction Laboratories | BD Transduction Laboratories: 611350; RRID: AB_398872 | IB 1:1000 |
| Antibody | Mouse monoclonal anti-clathrin | abcam | abcam: ab2731; RRID: AB_303256 | IF 1:200 |
| Antibody | Rabbit monoclonal anti-integrin β1 | abcam | abcam: ab52971; RRID: AB_870695 | IB 1:1000 |
| Antibody | Mouse monoclonal anti-transferrin receptor (H68.4) | Invitrogen | Invitrogen: 13-6800; RRID: AB_86623 | IB 1:1000 |
| Antibody | Mouse monoclonal anti-integrin β1 (TS2/16) | Invitrogen | Invitrogen: 14-0299-82; RRID: AB_1210468 | IF 1:500 |
| Antibody | Rat monoclonal anti-HA (3F10) | Roche | Roche: 11-867-423-001; RRID: AB_10094468 | IF 1:200 |
| Antibody | Alexa 488-, 594- or 633- secondaries | Molecular Probes | | IF 1:1000 |
| Antibody | Rabbit polyclonal anti-IRS-1 | PMID: 23478262 | | IP 1:200 |
| Recombinant DNA reagent | pFLAG-CMV-IRS-1 1-865 (plasmid) | This paper | | Vector: pFLAG-CMV; Insert: Rat IRS-1 1-865 |
| Recombinant DNA reagent | pFLAG-CMV-IRS-1 1-542 (plasmid) | This paper | | Vector: pFLAG-CMV; Insert: Rat IRS-1 1-542 |
| Recombinant DNA reagent | pFLAG-CMV-IRS-1 1-259 (plasmid) | This paper | | Vector: pFLAG-CMV; Insert: Rat IRS-1 1-259 |
| Recombinant DNA reagent | pFLAG-CMV-IRS-1 (plasmid) | This paper | | Vector: pFLAG-CMV; Insert: Rat IRS-1 full-length |
| Recombinant DNA reagent | pFLAG-CMV-IRS-2 (plasmid) | PMID: 21168390 | | Vector: pFLAG-CMV; Insert: human IRS-2 |
| Recombinant DNA reagent | pMXs-Puro-EGFP-IRS-1 (plasmid) | This paper | | Vector: pMXs-Puro; Insert: EGFP-IRS-1 wild-type |

*Continued on next page*

*Continued*

| Reagent type (species) or resource | Designation | Source or reference | Identifiers | Additional information |
|---|---|---|---|---|
| Recombinant DNA reagent | pMXs-Puro-EGFP-IRS-1 3YA (plasmid) | This paper | | Vector: pMXs-Puro; Insert: EGFP-IRS-1 3YA |
| Recombinant DNA reagent | pMXs-Puro-EGFP-IRS-1 ΔPTB (plasmid) | This paper | | Vector: pMXs-Puro; Insert: EGFP-IRS-1 DPTB |
| Recombinant DNA reagent | pMXs-Puro-EGFP (plasmid) | This paper | | Vector: pMXs-Puro; Insert: EGFP |
| Recombinant DNA reagent | pMXs-Puro-EGFP-IRS-2 (plasmid) | This paper | | Vector: pMXs-Puro; Insert: EGFP-rat IRS-2 |
| Recombinant DNA reagent | pIGF-IR-EGFP (plasmid) | This paper | | Vector: pEGFP-N1; Insert: human IGF-IR |
| Recombinant DNA reagent | pMXs-Puro-IGF-IR-FLAG (plasmid) | This paper | | Vector: pMXs-Puro; Insert: IGF-IR-FLAG |
| Recombinant DNA reagent | pMXs-Puro-IGF-IR-EGFP (plasmid) | This paper | | Vector: pMXs-Puro; Insert: IGF-IR-EGFP |
| Recombinant DNA reagent | pMXs-Puro-IGF-IR-HA-EGFP (plasmid) | This paper | | Vector: pMXs-Puro; Insert: IGF-IR-HA-EGFP |
| Recombinant DNA reagent | pMXs-Puro-integrin β1 (plasmid) | This paper | | Vector: pMXs-Puro; Insert: human integrin b1 |
| Recombinant DNA reagent | EGFR-GFP (plasmid) | Addgene | Addgene: 32751 | |
| Recombinant DNA reagent | pσ2-mRFP (plasmid) | This paper | | Vector: pCS2-mRFP4; Insert: rat s2 subunit |
| Recombinant DNA reagent | pmRFP-C1 (plasmid) | This paper | | |
| Recombinant DNA reagent | pmRFP-IRS-1 (plasmid) | This paper | | Vector: pmRFP-C1; Insert: rat IRS-1 |
| Recombinant DNA reagent | pGEX-μ1 (plasmid) | PMID: 23478262 | | Vector: pGEX-5X-3; Insert: mouse m1 |
| Recombinant DNA reagent | pGEX-μ2 (plasmid) | This paper | | Vector: pGEX-5X-3; Insert: mouse m2 |
| Recombinant DNA reagent | pGEX-C-μ2 (plasmid) | This paper | | Vector: pGEX-5X-3; Insert: mouse m2 C-terminal domain |
| Recombinant DNA reagent | pET15b-C-μ2 (plasmid) | This paper | | Vector: pET15b; Insert: rat m2 C-terminal domain |
| Recombinant DNA reagent | pLV-hU6-EF1a-green | Biosettia | Biosettia: SORT-B05 | |
| Recombinant DNA reagent | pCAG-HIVgp | RIKEN | RDB04394 | |
| Recombinant DNA reagent | pCMV-VSV-G-RSV-Rev | RIKEN | REB04393 | |
| Sequence-based reagent | siRNA targeting clathrin #1 | RNAi Corp. | | 5'-GUAUGCCUCUGAAUCGAAAGA-3' |
| Sequence-based reagent | siRNA targeting clathrin #2 | RNAi Corp. | | 5'-CAGAAGAAUCGACGUUAUUUU-3' |
| Sequence-based reagent | siRNA targeting μ2 #1 | RNAi Corp. | | 5'-CGAAGUGGCAUUUACGAAACC-3' |
| Sequence-based reagent | siRNA targeting μ2 #2 | RNAi Corp. | | 5'-CUGCUUUGGGAUAGUAUGAGC-3' |
| Sequence-based reagent | siRNA targeting IRS-1 #1 | RNAi Corp. | | 5'-CAAUGAGUGUGCAUAAACUUC-3' |
| Sequence-based reagent | siRNA targeting IRS-1 #2 | RNAi Corp. | | 5'-GCCUCGAAAGGUAGACACAGC-3' |

*Continued on next page*

*Continued*

| Reagent type (species) or resource | Designation | Source or reference | Identifiers | Additional information |
|---|---|---|---|---|
| Sequence-based reagent | siRNA targeting μ1 | RNAi Corp. | | 5'-CAGACGGAGAAUUCGAA CUCA-3' |
| Sequence-based reagent | Non-targeting control siRNA | RNAi Corp. | | 5'-GUACCGCACGUCAUUCG UAUC-3' |
| Sequence-based reagent | shRNA targeting LacZ | Invitrogen | | 5'-GCTACACAAATCAGCG ATTT-3'(targeting sequence) |
| Sequence-based reagent | shRNA targeting IRS-1 #5 | Invitrogen | | 5'-GCAGGCACCATCTCAAC AATCC-3'(targeting sequence) |
| Sequence-based reagent | shRNA targeting IRS-1 #6 | Invitrogen | | 5'-GAGAATATGTGAATATTG AATC-3'(targeting sequence) |
| Sequence-based reagent | Fbxo32-qPCR forward primer | Invitrogen | | ACTTCTCGACTGCCATCCTG |
| Sequence-based reagent | Fbxo32-qPCR reverse primer | Invitrogen | | TCTTTTGGGCGATGCCACTC |
| Sequence-based reagent | Trim63-qPCR forward primer | Invitrogen | | GGGAACGACCGAGTTCAGAC |
| Sequence-based reagent | Trim63-qPCR reverse primer | Invitrogen | | GCGTCAAACTTGTGGCTCAG |
| Sequence-based reagent | Fbxo30-qPCR forward primer | Invitrogen | | TGCAGTGGGGGAAAAAGAAGT |
| Sequence-based reagent | Fbxo30-qPCR reverse primer | Invitrogen | | TGCAGTACTGAATCGCCACA |
| Sequence-based reagent | Fbxo21-qPCR forward primer | Invitrogen | | ACTCCATCGGGCTCGTTATG |
| Sequence-based reagent | Fbxo21-qPCR reverse primer | Invitrogen | | TGTTTCGGATCCACTCGTGC |
| Sequence-based reagent | Map1lc3b-qPCR forward primer | Invitrogen | | GCCGGAGCTTCGAACAAAGA |
| Sequence-based reagent | Map1lc3b-qPCR reverse primer | Invitrogen | | GCTTCTCACCCTTGTATCGC |
| Sequence-based reagent | Gabarapl1-qPCR forward primer | Invitrogen | | ACAACACTATCCCTCCCACC |
| Sequence-based reagent | Gabarapl1-qPCR reverse primer | Invitrogen | | GCTTCTGCCTCATTTCCCGTA |
| Sequence-based reagent | Rn18s-qPCR forward primer | Invitrogen | | TCCCAGTAAGTGCGGGTCATA |
| Sequence-based reagent | Rn18s-qPCR reverse primer | Invitrogen | | CGAGGGCCTCACTAAACCATC |
| Peptide, recombinant protein | GST-μ1 | PMID: 23478262 | | GST-tagged mouse m1 |
| Peptide, recombinant protein | GST-μ2 | This study | | GST-tagged mouse m2 |
| Peptide, recombinant protein | GST-C-μ2 | This study | | GST-tagged mouse m2 C-terminal domain |
| Peptide, recombinant protein | His-C-μ2 | This study | | 6×His-tagged rat m2 C-terminal domain |
| Peptide, recombinant protein | GY(608)MPMSPG-IRS-1 peptide | Toray Research Center, Inc. | | Used for co-crystalization |
| Peptide, recombinant protein | DY(628)MPMSPK-IRS-1 peptide | Toray Research Center, Inc. | | Used for co-crystalization |
| Peptide, recombinant protein | GY(658)MMMSPS-IRS-1 peptide | Toray Research Center, Inc. | | Used for co-crystalization |

*Continued on next page*

*Continued*

| Reagent type (species) or resource | Designation | Source or reference | Identifiers | Additional information |
|---|---|---|---|---|
| Peptide, recombinant protein | recombinant human IGF-I | Astellas Pharma Inc. | | A kind gift from T. Ohkuma, Astellas Pharma Inc. |
| Peptide, recombinant protein | recombinant human EGF | Thermo Fisher Scientific | Thermo Fisher Scientific: PHG0315 | |
| Chemical compound, drug | Lipofectamine LTX | Invitrogen | Invitrogen: 15338100 | |
| Chemical compound, drug | Lipofectamine RNAiMAX | Invitrogen | Invitrogen: 13778075 | |
| Chemical compound, drug | leupeptin | PEPTIDE INSTITUTE, INC. | PEPTIDE INSTITUTE: 4041 | |
| Chemical compound, drug | pepstatin A | Sigma-Aldrich | Sigma-Aldrich: P5318-5MG | |
| Chemical compound, drug | Torin1 | Cayman Chemical | Cayman Chemical: 10997 | |
| Chemical compound, drug | rapamycin | Sigma-Aldrich | Sigma-Aldrich: 37094-10MG | |
| Chemical compound, drug | primaquine bisphosphate | Sigma-Aldrich | Sigma-Aldrich: 160393-1G | |
| Chemical compound, drug | cycloheximide | nacalai tesque | nacalai tesque: 06741-04 | |
| Chemical compound, drug | EZ-Link NHS-LC-Biotin | Pierce | Pierce: 21336 | |
| Chemical compound, drug | Biotin-SS-Sulfo-OSu | Dojindo | Dojindo: B572 | |
| Chemical compound, drug | LysoTracker Red DND-99 | Molecular Probes | Molecular Probes: L7528 | |
| Chemical compound, drug | Transferrin from human serum, Alexa Fluor 546 conjugate | Molecular Probes | Molecular Probes: T23364 | |
| Chemical compound, drug | Hoechst 33342 | Molecular Probes | Molecular Probes: H3570 | |
| Chemical compound, drug | ReverTra Ace qPCR Master Mix | TOYOBO | TOYOBO: FSQ-201 | |
| Chemical compound, drug | THUNDERBIRD SYBR qPCR Mix | TOYOBO | TOYOBO: QPS-201 | |
| Chemical compound, drug | cOmplete EDTA-free protease inhibitor cocktail | Roche | Roche: 11873580001 | |
| Software, algorithm | Fiji | PMID: 22743772 | RRID: SCR_002285 | |
| Software, algorithm | HKL2000 | PMID: 27754618 | | |
| Software, algorithm | CCP4 suite | PMID: 21460441 | RRID: SCR_007255 | |
| Software, algorithm | MOLREP | doi:10.1107/S0021889897006766 | | |
| Software, algorithm | REFMAC5 | PMID: 15299926 | RRID: SCR_014225 | |
| software, algorithm | PHENIX | PMID: 20124702 | RRID: SCR_014224 | |
| Software, algorithm | COOT | PMID: 15572765 | RRID: SCR_014222 | |
| Software, algorithm | PyMOL | The PyMOL Molecular Graphics System | RRID: SCR_000305 | |
| Other | Lenti-X Concentrator | Clontech | Clonetech: 631231 | |
| Other | Glutathione Sepharose 4B | GE Healthcare | GE Healthcare: 17075601 | |
| Other | Protein G Seharose Fast Flow | GE Healthcare | GE Healthcare: 17061801 | |

*Continued*

| Reagent type (species) or resource | Designation | Source or reference | Identifiers | Additional information |
|---|---|---|---|---|
| Other | Streptavidin Agarose | Pierce | Pierce: 20347 | |
| Other | HisTrap HP column | GE Healthcare | GE Healthcare: 17524801 | |
| Other | HiTrap SP HP column | GE Healthcare | GE Healthcare: 17115101 | |
| Other | HiLoad 16/60 Superdex200 column | GE Healthcare | GE Healthcare: 17-1069-01 | |

## Antibodies

Anti-phospho-IGF-IRβ (Tyr1131) antibody (3021), anti-phospho-IGF-IRβ (Tyr980) antibody (4568), anti-phospho-IGF-IRβ (Tyr1316) antibody (6113), anti-IGF-IRβ antibody (9750; for immunofluorescence staining), anti-Akt antibody (9272), anti-phospho-Akt (Thr308) antibody (9275), anti-phospho-Akt (Ser473) antibody (9271), anti-phospho-p70 S6K (Thr389) antibody (9234), anti-phospho-FoxO1 (Thr24)/FoxO3a (Thr32) antibody (9464), anti-phospho-FoxO1 (Ser256) antibody (9461), and anti-FoxO1 antibody (2880) were purchased from Cell Signaling Technology (Tokyo, Japan). Anti-IGF-IRα antibody (sc-712), anti-IGF-IRβ antibody (sc-713; for immunoblotting and immunoprecipitation), anti-IRS-2 antibody (sc-8299), anti-clathrin HC antibody (sc-12734; for immunoblotting), anti-α-adaptin antibody (sc-17771), anti-γ-adaptin antibody (sc-10763), anti-p70 S6K antibody (sc-230), anti-HSP90 antibody (sc-7947), anti-ubiquitin antibody (sc-8017) and anti-GFP antibody (sc-9996) were purchased from Santa Cruz Biotechnology (Santa Cruz, CA). Anti-FLAG M2 antibody, anti-α-tubulin antibody (DM1A), and anti-phospho-Tyr antibody (4G10) were purchased from Sigma-Aldrich (Tokyo, Japan). Anti-IRS-1 antibody (06–248), anti-myosin heavy chain (05–716) antibody, anti-Myc antibody (05–419), and anti-p85 PI3-kinase antibody (06–195) were purchased from Upstate (Lake Placid, NY). Anti-μ2 antibody (611350) was purchased from BD Biosciences (Tokyo, Japan). Anti-clathrin antibody (ab2731; for immunofluorescence staining), and anti-integrin β1 antibody (ab52971) were purchased from abcam (Tokyo, Japan). Anti-transferrin receptor antibody (H68.4) and anti-integrin β1 antibody (TS2/16) were purchased from Invitrogen (Tokyo, Japan). Anti-HA antibody (3F10) was purchased from Roche (Tokyo, Japan). IRS-1 polyclonal antibody for immunoprecipitation was raised in rabbit as previously described (*Yoneyama et al., 2013*).

## Cell culture and transfection

L6 and HEK293T cells were cultured as previously described (*Yoneyama et al., 2013*). The differentiation of L6 cells was induced as previously described (*Hakuno et al., 2011*). PLAT-E cells (provided by T. Kitamura, The University of Tokyo, Tokyo, JAPAN) were cultured for retrovirus packaging as previously described (*Yoneyama et al., 2013*). We tested each cell line for mycoplasma contamination and confirmed its absence using PCR Mycoplasma Test Kit I/C (PromoKine, Heidelberg, Germany) before experiments.

The transfection of expression plasmids was performed by using polyethylenimine (PEI) for HEK293T cells as previously described (*Lanzerstorfer et al., 2015*), or by using Lipofectamine LTX (Invitrogen) for L6 cells. For RNA interference (RNAi), the cells were transfected with the following siRNAs (RNAi Corp., Tokyo, Japan) by using Lipofectamine RNAiMAX (Invitrogen) according to the manufacturer's instructions: clathrin (#1), 5'-GUAUGCCUCUGAAUCGAAAGA-3'; clathrin (#2), 5'-CAGAAGAAUCGACGUUAUUUU-3'; μ2 (#1), 5'-CGAAGUGGCAUUUACGAAACC-3'; μ2 (#2), 5'-CUGCUUUGGGAUAGUAUGAGC-3'; IRS-1 (#1), 5'-CAAUGAGUGUGCAUAAACUUC-3'; IRS-1 (#2), 5'-GCCUCGAAAGGUAGACACAGC-3'; μ1, 5'-CAGACGGAGAAUUCGAACUCA-3'; non-targeting control (Ctrl, 5'-GUACCGCACGUCAUUCGUAUC-3'.

## Expression plasmids

A series of IRS-1 deletion mutants (amino acid residues 1–865, 1–542, 1–259 and full-length of rat IRS-1) were cloned into pFLAG-CMV vector. The full-length of IRS-1 was also cloned into pmRFP-C1 vector. EGFP-fused IRS-1 and 3YA (Y608A/Y628A/Y658A) (*Yoneyama et al., 2013*) were cloned into pMXs-Puro vector (provided by T. Kitamura, The University of Tokyo, Tokyo, JAPAN). FLAG-fused IRS-1 was also cloned from pFLAG-CMV-IRS-1 into pMXs-Puro. The construction of pFLAG-CMV-

IRS-2 was described previously (*Fukushima et al., 2011*). EGFP-fused IRS-2 was also cloned from pEGFP-IRS-2 (*Lanzerstorfer et al., 2015*) into pMXs-Puro. Full-length IGF-IR was cloned into pEGFP-N1 to generate the construct of IGF-IR fused with EGFP at its C-terminus. IGF-IR-EGFP and IGF-IR-FLAG (*Fukushima et al., 2012*) were then cloned into pMXs-Puro. To generate the double-tagged IGF-IR construct (IGF-IR-HA-EGFP), the fragment encoding the α subunit attached to the HA epitope (α + HA) and the fragment encoding the β subunit attached to the HA epitope (β + HA) were prepared by PCR with independent primer sets as follows: for α + HA, 5'-CTCAAGCTTCGAATTCATGAAGTCTGGCTCCGGA-3' and 5'-TGGAACATCGTATGGGTACATGGTggccacttgcatgacatctctc-3'; for β + HA, 5'-CCATACGATGTTCCAGATTACGCTaacaccaccatgtccagccgaa-3' and 5'-GGCGACCGGTGGATCCGCGCAGGTCGAAGACTGGGGCA-3'. The two fragments were cloned into pEGFP-N1 by using In-Fusion Cloning HD Kit (TAKARA). The IGF-IR-HA-EGFP was then cloned into pMXs-Puro. The cDNA of human integrin β1 was cloned into pMXs-Puro. The expression plasmid of EGFR fused with EGFP was purchased from Addgene (#32751). The cDNA encoding rat σ2 subunit of the AP2 complex was obtained from pACT2-σ2 (provided by H. Ohno, RIKEN, Kanagawa, Japan), and cloned into pCS2-mRFP4 (provided by M. Taira, The University of Tokyo, Tokyo, Japan). The cDNA encoding human PTP1B was cloned into pCMV5-Myc vector, and the D181A mutation was introduced by site-directed mutagenesis. Construction of pGEX-µ1 was described previously (*Yoneyama et al., 2013*). The full-length cDNA of mouse µ2 was obtained from pcDNA-µ2 (provided by H. Ohno, RIKEN, Kanagawa, Japan) and cloned into pGEX-5X-3. To generate the construct for the recombinant C-terminal region of rat µ2 fused with His-tag, the region corresponding to amino acid residues 158–435 was cloned by RT-PCR using total RNA isolated from L6 cells and subcloned into pET15b.

## Retrovirus production and generation of stable cell lines

Retrovirus production and retrovirus transduction in L6 cells were performed as described previously (*Yoneyama et al., 2013*). Briefly, PLAT-E cells were transiently transfected with pMXs-Puro vectors by using PEI reagent, and the medium containing retrovirus was collected. L6 cells were incubated with the virus-containing medium supplemented with 2 µg/ml of polybrene. Uninfected cells were removed by puromycin selection. L6 cells expressing EGFP-fused constructs were further isolated using a FACSAria II cell sorter (BD Biosciences) as EGFP-positive cells.

## Lentivirus production and shRNA expression in L6 myotubes

For lentiviral RNAi, shRNA sequences against IRS-1 were cloned into pLV-hU6-EF1a-green (Biosettia, San Diego, CA) according to the manufacturer's instructions. The shRNAs used in this study comprised the following sequences: shLacZ, 5'-GCTACACAAATCAGCGATTT-3'; shIRS-1_5, 5'-GCAGGCACCATCTCAACAATCC-3'; shIRS-1_6, 5'-GAGAATATGTGAATATTGAATC-3'. HEK293T cells were transiently transfected with pLV-hU6-EF1a-green vectors together with pCAG-HIVgp and pCMV-VSV-G-RSV-Rev (provided by RIKEN BRC, Ibaraki, Japan) by using PEI reagent, and the medium containing lentivirus was collected followed by concentration with Lenti-X Concentrator (Clontech, Fremont, CA) to achieve high titer virus. The virus titer was evaluated by GFP fluorescence expressed from pLV-hU6-EF1a-green vector in L6 myoblasts infected with serially diluted virus-containing medium. Lentiviral infection was conducted on the second day of differentiation. The virus-containing medium supplemented with 8 µg/ml of polybrene was added into L6 myotube culture, and the culture plates were spun at 1200 *g* for 1 hr at room temperature to increase the infection efficiency. After incubation for 1 day, differentiation medium was replaced, and the myotubes were cultured for additional 5 days.

## In vitro pull-down assay

Purification of GST-fused proteins from *E. coli* BL21 and pull-down assays were performed as described previously (*Yoneyama et al., 2013*). Briefly, lysates of L6 cells or HEK293T cells expressing GFP-IRS-1 mutants were incubated with purified GST-fused proteins bound to Glutathione Sepharose 4B (GE Healthcare, Tokyo, Japan). Bound proteins were analyzed by immunoblotting with the indicated antibody.

## Cell stimulation and immunoblotting

Recombinant human IGF-I was kindly donated by T Ohkuma (Astellas Pharma Inc., Tokyo, Japan). Recombinant human EGF was purchased from Thermo Fisher. Prior to ligand stimulation, the cells were serum-starved for 12 hr in Dulbecco's modified Eagle's medium (DMEM) supplemented with 0.1% bovine serum albumin (BSA), and then treated with the ligand (100 nM IGF-I or 100 nM EGF) for the indicated time. When needed, cells were preincubated for 30 min with chemical inhibitors at the following concentrations: 250 µg/ml leupeptin (PEPTIDE INSTITUTE, INC., Osaka, Japan), 10 µg/ml pepstatin A (Sigma-Aldrich), 100 nM Torin1 (Cayman Chemical), 100 nM rapamycin (Sigma-Aldrich), 0.1 mM primaquine (Sigma-Aldrich), and 10 µg/ml cycloheximide (Nacalai Tesque, Inc., Kyoto, Japan).

After the treatment, the extraction of cell lysate and immunoblotting were performed as described previously (*Yoneyama et al., 2013*). Densitometry was performed in the linear phase of the exposure by using ImageJ software. The results were expressed as the percent of max, which corresponds to the highest value of phosphorylation among the time course experiments of control cells. Values represent means ±SEM from at least three independent experiments.

## Immunoprecipitation

After the treatment of inhibitors and ligands, cells were rinsed once with ice-cold PBS and then lysed in lysis buffer (25 mM Tris-HCl, pH 7.4, 150 mM NaCl, 1 mM EDTA, 10% glycerol, 1% Triton X-100, 100 Kallikrein inhibitor units [KIU]/ml aprotinin, 20 µg/ml phenylmethylsulfonyl fluoride [PMSF], 10 µg/ml leupeptin, 5 µg/ml pepstatin A, 500 µM $Na_3VO_4$, and 10 mg/ml $p$-nitrophenyl phosphate [PNPP]). After brief sonication, the clear supernatant was obtained by centrifugation at 15,000 $g$ for 15 min at 4°C. For immunoprecipitation of IRS-1 or IGF-IR, the lysates were incubated with anti-IRS-1 antibody or anti-IGF-IRβ antibody (Santa Cruz) overnight at 4°C, and further incubated in the presence of Protein G Sepharose beads (GE healthcare). For immunoprecipitation of FLAG fusion proteins, the lysates were incubated with anti-FLAG M2 affinity gel beads (Sigma-Aldrich) for 2 hr. Immunoprecipitates were collected by centrifugation and washed three times with lysis buffer, and then proteins were eluted with Laemmli's sample buffer. Samples were analyzed by immunoblotting with the indicated antibodies.

## Surface biotinylation and internalization assay

Surface IGF-IR levels were measured as follows. L6 cells were treated with IGF-I for the indicated time, then placed on ice, washed three times with ice-cold PBS, and labeled for 30 min with Sulfo-NHS-LC-biotin (0.5 mg/ml; Pierce) in PBS at 4°C. Biotinylation was then quenched with 15 mM glycine in PBS. After washing the cells with PBS once, they were lysed in lysis buffer. After brief sonication, the supernatant was obtained by centrifugation at 15,000 $g$ for 15 min at 4°C. The cleared lysates were then incubated with Streptavidin agarose beads (Pierce, Tokyo, Japan) overnight at 4°C. The beads were washed three times with lysis buffer, and bound proteins were eluted with Laemmli's sample buffer. Samples were analyzed by immunoblotting with the indicated antibodies.

Internalization of IGF-IR was measured as follows. Serum-starved L6 cells were washed three times with cold PBS before incubation with 0.2 mg/ml Biotin-SS-Sulfo-OSu, a nonpermeable and reversible biotinylation reagent (Dojindo, Kumamoto, Japan), in PBS for 30 min at 4°C. After surface labeling, cells were washed twice with 15 mM glycine in PBS on ice, and transferred to 0.1% BSA in DMEM with or without the ligand to allow internalization. At the indicated times, cells were washed once with cold PBS and treated twice with 100 mM MesNa (50 mM Tris-HCl, pH 8.6, 100 mM NaCl, and 0.1% BSA), a nonpermeable reducing regent, for 15 min at 4°C to remove biotin. MesNa was quenched with 5 mg/ml iodoacetamide in PBS for 10 min at 4°C. After two cold PBS washes, cells were lysed followed by streptavidin pull-down as described above.

## Ubiquitination assay

Cells were washed with ice-cold PBS and then lysed with lysis buffer supplemented with 100 mM $N$-ethylmaleimide (NEM). The cleared lysates were subjected to immunoprecipitation with anti-FLAG M2 affinity gel beads. The immunoprecipitates were then washed three times with lysis buffer supplemented with 100 mM NEM, and heated in 50 mM Tris-HCl, pH 7.4, 150 mM NaCl, and 1% SDS at 98°C for 5 min to disrupt non-covalent protein-protein interactions. The supernatants diluted with

lysis buffer (1:10) were re-immunoprecipitated with anti-FLAG M2 affinity gel beads, and then subjected to SDS-PAGE. After transfer to PVDF membranes, the membranes were subjected to a denaturing treatment prior to blocking the primary antibody by incubation for 30 min at 4°C in 50 mM Tris-HCl, pH 7.5, 6M guanidine-HCl, and 5 mM 2-mercaptoethanol.

## Immunofluorescence staining

For confocal microscopy L6 cells were grown on coverslips. For TIRF microscopy the cells were grown on Glass Bottom Dish Hydro (MATSUNAMI, Osaka, Japan). In both cases, the cells were fixed for 20 min at room temperature in prewarmed 4% paraformaldehyde in PBS. The fixed cells were then washed three times with PBS and subsequently incubated for 5 min in 50 mM ammonium chloride in PBS. After washing three times with PBS, cells were permeabilized with 0.25% Triton X-100 in PBS at room temperature for 5 min. The cells were washed three times with PBS and then blocked for 1 hr at room temperature with BSA blocking buffer (3% BSA and 0.025% NaN$_3$ in PBS). Primary antibodies diluted in BSA blocking buffer were added overnight at 4°C. The samples were washed three times with PBS and incubated for 1 hr at room temperature in the solution of Alexa Fluor-conjugated secondary antibodies diluted in BSA blocking buffer. For LysoTracker experiments, Lyso-Tracker Red DND-99 (Life Technologies, Tokyo, Japan) was added to cells at the concentration of 50 nM 30 min prior to fixation. Fixed cells were stained with Hoechst 33342 (Molecular Probes, Tokyo, Japan) to visualize nuclei. Coverslips were mounted in Vectashield (Vector Laboratories, Burlingame, CA) for confocal microscopy. Fixed cells in glass bottom dishes were imaged in PBS for TIRF microscopy.

To chase surface IGF-IR, L6 cells stably expressing IGF-IR-HA-EGFP were serum-starved, washed three times with ice-cold Hank's Balanced Salt Solution (HBSS), and then incubated on ice for 1 hr with 2 µg/ml anti-HA antibody diluted in HBSS. After removing the excess antibody, cells were incubated in 0.1% BSA in DMEM with or without IGF-I at 37°C for different time periods. At each time point, non-permeabilized cells were either fixed to visualize the surface receptor or acid washed in an ice-cold buffer (100 mM glycine, 20 mM Mg(OAc)$_2$, and 50 mM KCl, pH 2.2) to strip surface-bound antibody. Cells were fixed and permeabilized to visualize the internalized receptor.

To examine endocytosis of transferrin, L6 cells were serum-starved for 30 min, and incubated with 25 µg/ml Alexa Fluor 546-conjugated transferrin (Invitrogen) for the indicated time. Surface-bound fraction was evaluated from the cells labeled with Alexa Fluor 546-conjugated transferrin at 4°C. The rate of uptake is expressed as internalized/surface-bound fluorescent intensity. To examine EGF-dependent internalization of EGFR, L6 cells transfected with pEGFR-EGFP plasmid were treated with 2 nM EGF for indicated time. To examine internalization of integrin β1, L6 cells stably expressing human integrin β1 were serum-starved and then labeled with anti-integrin integrin β1 antibody (TS2/16), which recognizes human integrin β1, for 30 min on ice. After removing the excess antibody, cells were incubated in 0.1% BSA in DMEM at 37°C. At each time point, cells were washed in ice-cold acid buffer to strip surface-bound antibody. Fixed cells were observed by confocal microscopy.

## Microscopy and image quantification

Confocal imaging of fixed and fluorescently stained samples was performed on an inverted Olympus FV1200 microscope. Appropriate excitation and emission wavelengths were configured by the instrument running FV10-ASW software, and emission signals in the different channels were collected in the sequential scan mode. TIRF imaging of fixed and fluorescently stained samples was performed on Leica AF6000LX total internal reflection (TIRF) microscopy equipped with a 100 × 1.46 NA oil-immersion objective and a Cascade II EMCCD camera (Roper, Tucson, AZ). Images were analyzed with Adobe Photoshop CC2017 and Fiji software. Live cell dual-color TIRF microscopy was carried out as described previously (*Lanzerstorfer et al., 2015*).

Quantifications were performed with Fiji software. Mean fluorescence levels in individual cells minus the background fluorescence were calculated and averaged. For colocalization analysis, background intensity was subtracted by median subtraction, the value of Mander's colocalization coefficient (MCC), which is one of the most widely accepted methods to measure colocalization of different markers (*Dunn et al., 2011*), was calculated by Fiji plugin in individual cells. The number of AP2-positive spots was determined as follows. Punctate structures were extracted using median

subtraction, and binary images were created. Small punctae less than 5 pixel$^2$ were removed, and the number of spots was counted using the morpheme analysis program.

Images of differentiated myotubes were obtained by BZ-9000 microscope (Keyence, Osaka, Japan). Myotube diameter was quantified by measuring a total of over 100 tube diameters from ten random fields using Fiji software.

## Quantitative RT-PCR analysis

Total RNA from L6 cells was extracted with TRIzol reagent (Invitrogen) from three independently collected cells. First-strand cDNA was synthesized with ReverTra Ace qPCR Master Mix (TOYOBO, Osaka, Japan). Quantitative PCR was performed with THUNDERBIRD SYBR qPCR Mix (TOYOBO) on an ABI StepOnePlus Real Time PCR System (Applied Biosystems). To normalize the relative expression, a standard curve was prepared for each gene for relative quantification, and the expression level of each gene was normalized to the *Rn18s* gene. Specific primers for atrophy-related genes were used: *Fbxo32* F: ACTTCTCGACTGCCATCCTG; *Fbxo32* R: TCTTTTGGGCGA TGCCACTC; *Trim63* F: GGGAACGACCGAGTTCAGAC; *Trim63* R: GCGTCAAACTTGTGGCTCAG; *Fbxo30* F: TGCAGTGGGGGGAAAAAGAAGT; *Fbxo30* R: TGCAGTACTGAATCGCCACA; *Fbxo21* F: ACTCCATCGGGCTCGTTATG; *Fbxo21* R: TGTTTCGGATCCACTCGTGC; *Map1lc3b* F: GCCGGAGC TTCGAACAAAGA; *Map1lc3b* R: GCTTCTCACCCTTGTATCGC; *Gabarapl1* F: ACAACACTATCCC TCCCACC; *Gabarapl1* R: GCTTCTGCCTCATTTCCCGTA; *Rn18s* F: TCCCAGTAAGTGCGGGTCATA; *Rn18s* R: CGAGGGCCTCACTAAACCATC.

## Yeast two-hybrid assay

Yeast two-hybrid assay using pAS-IRS-1 and pACT2-μ2 to assess the interaction between IRS-1 and μ2 was performed as described previously (*Hakuno et al., 2007*).

## Protein expression and purification

Construct of 6 × His tagged C-μ2 (rat μ2 amino acid residues 158–435) cloned into pET15b was transformed into an *E. coli* strain BL21-CodonPlus(DE3)-RIL (Agilent Technologies, Santa Clara, CA). Bacteria were grown in LB supplemented with ampicillin and chloramphenicol at 37°C to OD$_{600}$ of 0.7. Expression was induced with 0.1 mM isopropyl β-$_\mathrm{D}$-thiogalactopyranoside (IPTG) at 17°C overnight. The cells were harvested by centrifugation and homogenized with a sonicator in a buffer of 50 mM Tris-HCl (pH 8.0), 500 mM NaCl, 20 mM imidazole, 5% glycerol, and 0.1% Triton X-100 supplemented with cOmplete EDTA-free protease inhibitor cocktail (Roche). Insoluble material was removed by centrifugation. The protein was affinity-purified on HisTrap HP column (GE Healthcare). The His-tag was removed by cleavage of thrombin at room temperature for 4 hr. Thrombin-cleaved C-μ2 was further purified with HiTrap SP HP column (GE Healthcare), and uncleaved fusion protein was removed by passage through HisTrap HP column. The C-μ2 was finally purified by gel filtration on HiLoad 16/60 Superdex200 column equilibrated in a buffer of 10 mM HEPES-KOH (pH 7.5), 150 mM NaCl, and 2 mM dithiothreitol (DTT) for crystallization.

## Crystallization and structure determination

Three eight-residue peptides of IRS-1 were chemically synthesized with their sequences GY(608) MPMSPG, DY(628)MPMSPK and GY(658)MMMSPS, where the tyrosine residue in a YxxΦ motif is indicated with its residue number in parentheses (Toray Research Center, Inc., Tokyo, Japan). Hereafter, they are referred to as Y608 peptide, Y628 peptide, and Y658 peptide, respectively. The peptides were dissolved in 10 mM HEPES buffer (pH 7.5) containing 150 mM NaCl and 2 mM DTT. C-μ2 was mixed with each peptide in the molecular ratio of 1:10. Crystals of the Y608 peptide were grown by the sitting drop method at 293 K with the reservoir solution containing 1.4 M sodium formate, 50 mM nickel chloride and 100 mM sodium acetate (pH 6.0). Crystals of the Y628 and Y658 peptides were grown by the hanging drop method at 291 K with the reservoir solution containing 2.2–2.3 M sodium chloride, 400 mM sodium potassium phosphate, 10 mM DTT, 15% (v/v) glycerol and 100 mM MES (pH 6.5). Crystals were briefly soaked in well solution containing 20% (v/v) glycerol before flash-cooled in liquid nitrogen. Diffraction data were collected on BL26B2 at SPring-8, Harima, Japan, and processed using HKL2000 (*Otwinowski and Minor, 1997*) and the CCP4 suite (*Winn et al., 2011*). Molecular replacement was carried out with CCP4 program MOLREP

(**Vagin and Teplyakov, 1997**) using the μ2 subunit in the complex with EGFR internalization signal peptide (**Owen and Evans, 1998**) (PDB 1BW8) as the search model. Refinement was performed with REFMAC5 (**Murshudov et al., 1997**) and PHENIX (**Adams et al., 2010**), while model building was performed with COOT (**Emsley and Cowtan, 2004**). The N-terminal residue and residues 220–237 of C-μ2 were not modeled for the complexes of the Y628 and Y658 peptides. As for the complex with the Y608 peptide, it appeared that the region encompassing residues 219–260 underwent a conformational change where the electron density was not enough to precisely trace the structure. Residues 224–260 were not modeled except for a five-alanine strand which was placed as unconfirmed residues in a patch of visible electron density. Structural models in the figures were drawn using PyMOL (The PyMOL Molecular Graphics System, Schrödinger, LLC). Coordinates and structure factors of the three complexes have been deposited in the Protein Data Bank (PDB) with accession codes indicated in **Table 1**.

## Statistical analysis

Comparisons between two groups were performed using two-tailed, unpaired Student's $t$ test, whereas comparisons among more than two groups were analyzed by analysis of variance (ANOVA) and the Tukey *post hoc* test. p Values of $< 0.05$ were considered statistically significant.

## Acknowledgements

We thank Shinya Mimasu (The University of Tokyo) for initial crystallographic analysis and Shin Sato (RIKEN CLST) for help in biochemical analysis, and Takashi Minowa of NIMS MMS platform in 'Nanotechnology Platform Project', which is supported by the Ministry of Education, Culture, Sports, Sciences and Technology (MEXT), for microscopy. We also thank members of the Takahashi lab for valuable support and discussion, and Susan Hall (University of North Carolina) and Marc Tatar (Brown University) for critically reading the manuscript. We thank Ignacio Torres Alemán (Cajal Institute) and Leonie Reiger (University College Cork) for discussion and technical advices. This work was supported in part by Grants-in-Aid for the Japan Society for the Promotion of Science (JSPS) Fellows and for Young Scientists (B) #15K18766 from JSPS to YY; the Targeted Proteins Research Program (TPRP) from MEXT and the Platform Project for Supporting in Drug Discovery and Life Science Research (Platform for Drug Discovery, Informatics, and Structural Life Science) from MEXT and Japan Agency for Medical Research and development (AMED) to SY; the Austrian Research Promotion Agency (FFG; project number 850681), the University of Applied Sciences Upper Austria Basic Funding initiative (project GlucoSTAR) and the Center for Technological Innovation in Medicine (TIMed Center) to JW; Grant-in-Aid for Scientific Research (S) #25221204 and Core-to-core program A A Advanced Research Networks from JSPS to S-IT.

## Additional information

### Funding

| Funder | Grant reference number | Author |
|---|---|---|
| Japan Society for the Promotion of Science | 15K18766 | Yosuke Yoneyama |
| Ministry of Education, Culture, Sports, Science, and Technology | The Targeted Proteins Research Program (TPRP) | Shigeyuki Yokoyama |
| Japan Agency for Medical Research and Development | | Shigeyuki Yokoyama |
| Japan Agency for Medical Research and Development and Ministry of Education, Culture, Sports, Science, and Technology | Platform Project for Supporting in Drug Discovery and Life Science Research | Shigeyuki Yokoyama |
| Austrian Research Promotion Agency (FFG) | 850681 | Julian Weghuber |

| Japan Society for the Promotion of Science | | Shin-Ichiro Takahashi |
| Center for Technological Innovation in Medicine, TIMed Center | | Julian Weghuber |
| University of Applied Sciences Upper Austria and the Center for Technological Innovation in Medicine (TIMed Center) | Project GlucoSTAR | Julian Weghuber |

The funders had no role in study design, data collection and interpretation, or the decision to submit the work for publication.

## Author contributions

Yosuke Yoneyama, Conceptualization, Data curation, Supervision, Funding acquisition, Validation, Investigation, Writing—original draft, Writing—review and editing; Peter Lanzerstorfer, Conceptualization, Data curation, Formal analysis, Funding acquisition, Validation, Investigation, Methodology, Writing—original draft, Writing—review and editing; Hideaki Niwa, Data curation, Formal analysis, Investigation, Methodology, Writing—original draft; Takashi Umehara, Conceptualization, Data curation, Formal analysis, Investigation, Methodology, Writing—original draft; Takashi Shibano, Conceptualization, Data curation, Formal analysis, Investigation, Writing—original draft; Shigeyuki Yokoyama, Resources, Data curation, Funding acquisition, Methodology; Kazuhiro Chida, Resources, Funding acquisition, Methodology; Julian Weghuber, Conceptualization, Resources, Data curation, Formal analysis, Investigation, Methodology; Fumihiko Hakuno, Conceptualization, Data curation, Formal analysis, Supervision, Investigation, Methodology, Writing—original draft, Writing—review and editing; Shin-Ichiro Takahashi, Conceptualization, Data curation, Formal analysis, Supervision, Funding acquisition, Investigation, Writing—original draft, Writing—review and editing

## Author ORCIDs

Yosuke Yoneyama http://orcid.org/0000-0002-9170-2846
Shin-Ichiro Takahashi http://orcid.org/0000-0002-2323-2010

## Decision letter and Author response

Decision letter https://doi.org/10.7554/eLife.32893.026
Author response https://doi.org/10.7554/eLife.32893.027

# Additional files

## Supplementary files

• Transparent reporting form
DOI: https://doi.org/10.7554/eLife.32893.024

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
