## [Decision Letter]

Thank you for submitting your article "IRS-1 acts as an endocytic regulator of IGF-I receptor to facilitate sustained IGF signaling" for consideration by *eLife*. Your article has been reviewed by three peer reviewers, one of whom is a member of our Board of Reviewing Editors and the evaluation has been overseen by Philip Cole as the Senior Editor. The reviewers have opted to remain anonymous.

The reviewers have discussed the reviews with one another and the Reviewing Editor has drafted this decision to help you prepare a revised submission.

Summary:

This is an interesting study that describes a role for IRS-1 as a ligand for AP-2 adapter proteins that mediate recruitment of cell surface proteins to clathrin coated pits for endocytosis. The authors propose that IRS-1 binding prevents AP-2 recognition of IGF-1R by competition and therefore suppresses IGF-1R internalization to enable sustained IGF-1 signaling. Several lines of evidence are presented to support these conclusions. Nevertheless, there are several issues that require clarification.

Essential revisions:

1) The biotinylation assays for cell surface IGF-1R were performed following incubation with IGF-1 (Figure 1). It is therefore not clear that the maintenance of cell surface IGF-1R that is observed is not caused by recycling of internalized receptors rather than the failure to internalize. Moreover, the interpretation of the loss of pTyr from the cell surface is unclear in this assay. The authors appear to assume that this reflects internalization of cell surface pTyr-labeled receptors, but why could this not be caused by dephosphorylation? Direct assays for receptor internalization are required for the authors to draw conclusions concerning IGF-1R internalization and the role of IRS-1.

2) The authors' estimations of the rate of IGF-IR endocytosis in L6 cells is unusually long. A small decrease in the amount of pIGF-IR is observed only after 3 hours of continuous stimulation of cells with IGF1, and the total surface receptor does not change at all. This is highly unusual for a cargo internalized by CME. Technically, because surface biotinylation is used to monitor plasma membrane receptor levels at "hours" time-scale, rates of internalization, recycling, degradation and insertion of newly synthesized receptors are all contribute to the apparent rate of down-regulation in these measurements. Therefore, whether manipulations with IRS1 levels affect the AP-2 mediated CME of the receptor is not directly demonstrated. Notably, receptors are shown to accumulate in large clathrin structures on the cell-bottom membrane. These clathrin plaques are "endocytosis-passive" in many types of cells. Localization in these structures may not be as a "positive" endocytic event and is not a useful correlate of the internalization rate.

3) The authors propose that phosphotyrosine prevents the binding of the IRS-1 Yxxø motifs to AP-2. However, treatment of cells with IGF-1 does not inhibit the co-immunoprecipitation of IRS-1 with AP-2. The authors argue that this is because of low stoichiometry tyrosine phosphorylation of IRS-1. This could be tested by co-immunoprecipitation analysis.

4) The interpretation of the IRS-1 over-expression experiments is unclear. If IRS-1 binds AP-2, this should inhibit the internalization of many AP-2 cargos, but this was not observed – e.g. transferrin receptor. The authors argue that this could be because IRS-1 has some special localization within the cell; would this also be true for over-expressed IRS-1? If IRS-1 is an AP-2 competitor, why does it not inhibit internalization of other AP-2 dependent proteins when over-expressed? Moreover, if the competition for AP-2 is restricted to the local environment of the IGF-1R, does this mean that it does not affect other cell surface proteins that signal through IRS-1 signals (e.g. integrins) and if so, why not? Might this mechanism also affect transmembrane tyrosine phosphatases that dephosphorylate IGF-1R?

5) The kinetics of endocytosis of pIGF-IR and signaling are on different scales. The effects of the depletion or overexpression of IRS1 are evident on pAkt only after 6-12 hours (Figure 7) or later. The effects of IRS1 level alterations can be alternatively interpreted by the increase or decrease of the amount of phosphoIRS1 capable of engaging PI3K. The lack of the effect of the blockade of the receptor endocytosis by mu2 depletion of Akt activity (Figure 7—figure supplement 1D and 1E) is puzzling. The authors' interpretation of this data is that slow endocytosis is not sufficient in order to prolong Akt activity, and that the presence of high levels of IRS1 is also required. Such interpretation is difficult to reconcile with the model.

---

## [Author Response]

Essential revisions:1) The biotinylation assays for cell surface IGF-1R were performed following incubation with IGF-1 (Figure 1). It is therefore not clear that the maintenance of cell surface IGF-1R that is observed is not caused by recycling of internalized receptors rather than the failure to internalize. Moreover, the interpretation of the loss of pTyr from the cell surface is unclear in this assay. The authors appear to assume that this reflects internalization of cell surface pTyr-labeled receptors, but why could this not be caused by dephosphorylation? Direct assays for receptor internalization are required for the authors to draw conclusions concerning IGF-1R internalization and the role of IRS-1.

We have added several key experiments that address the reviewer’s points and strengthen our conclusion. We have addressed the potential reasons for the maintenance of cell surface IGF-IR in IGF-I-stimulated cells, including the recycling of internalized receptor and receptor dephosphorylation. To inhibit the recycling, we used primaquine which has reportedly inhibited the recycling of transferrin receptor and receptor tyrosine kinases. Cells were pre-treated with primaquine followed by IGF-I stimulation, and then surface-biotinylated to measure the time-dependent changes in surface IGF-IR (Figure 3C in the revised manuscript). The results showed that surface levels of transferrin receptor were reduced by primaquine treatment. Surface IGF-IR levels were reduced by IGF-I treatment within 1 hour, and phospho-IGF-IR levels followed this change (Figure 3C in the revised manuscript). These observations indicate that the recycling contributes to the apparent surface maintenance of IGF-IR, which had been observed in Figure 2A and 2B of the original manuscript. The data also support the notion that IGF-I triggers ligand-dependent endocytosis of IGF-IR in L6 cells.

In response to the reviewer’s question regarding the dephosphorylation of IGF-IR, we focused on PTP1B, an endoplasmic reticulum-resident protein tyrosine phosphatase, which has reportedly downregulated IGF-IR by dephosphorylation (Buckley et al., 2002). The PTP1B substrate-trapping mutant (D181A) binds to but cannot dephosphorylate its substrate. Phosphorylation levels of IGF-IR observed 1 hour after IGF-I treatment and the subsequent reduction in the later period (6 hours) were comparable between PTP1B D181A-expressing and non-expressing cells as revealed by immunofluorescence (Figure 3—figure supplement 2B in the revised manuscript). Although we could not rule out the involvement of other tyrosine phosphatases targeting IGF-IR, our observation indicates a negligible role of PTP1B in the down-regulation of phospho-IGF-IR in our assay.

In response to the reviewer’s suggestion, we have directly measured the internalized IGF-IR via two different approaches. First, we performed the internalization assay using the non-permeable and cleavable biotinylation reagent (please see details in Materials and methods section). In this assay, cells were surface-biotinylated followed by incubation with or without IGF-I. Before lysis, residual surface biotin was stripped off by treating cells with a non-permeable reducing agent MesNa, which cleaves the disulfide-coupled biotin. The biotinylated proteins that were internalized from the cell surface were isolated by streptavidin pull-down. It revealed that internalized IGF-IR was detected within 15 minutes after surface biotinylation (Figure 3—figure supplement 3A in the revised manuscript).

To complement this result, we developed a double-tagged IGF-IR construct (IGF-IR-HA-EGFP) that contains an extracellular HA-tag and intracellular EGFP as a second approach. Internalization can be directly monitored by following uptake of anti-HA antibody added to the media prior to ligand treatment (Figure 3—figure supplement 3B, C in the revised manuscript). In L6 cells stably expressing IGF-IR-HA-EGFP, internalized fraction of the double-tagged IGF-R was detected within 15 to 60 minutes under both IGF-I-stimulated and non-stimulated conditions (Figure 3—figure supplement 3D, E in the revised manuscript). The internalization of IGF-IR observed in the non-stimulated state was not affected by knockdown of AP2 (Figure 3—figure supplement 3F in the revised manuscript), indicating that the basal endocytosis of IGF-IR is not dependent on AP2. In contrast, phospho-IGF-IR was predominantly localized to the cell surface and did not overlap with internalized IGF-IR (HA-positive) within 1 hour in the ligand-stimulated cells while IGF-I did not affect the apparent uptake of anti-HA antibody within 1 hour (Figure 3—figure supplement 3D). At the later period (6 hours), phospho-IGF-IR was detected in LysoTracker-positive compartments (Figure 3—figure supplement 1B, left in the revised manuscript). More importantly, the phospho-IGF-IR targeting to lysosomes was abolished by knockdown of AP2 (Figure 3—figure supplement 1B, right; Figure 3—figure supplement 1C n the revised manuscript). These observations therefore demonstrate that IGF-IR undergoes endocytosis in L6 cells both in ligand-stimulated and non-stimulated conditions, and in particular that ligand-activated IGF-IR is internalized in an AP2-dependent manner.

2) The authors' estimations of the rate of IGF-IR endocytosis in L6 cells is unusually long. A small decrease in the amount of pIGF-IR is observed only after 3 hours of continuous stimulation of cells with IGF1, and the total surface receptor does not change at all. This is highly unusual for a cargo internalized by CME. Technically, because surface biotinylation is used to monitor plasma membrane receptor levels at "hours" time-scale, rates of internalization, recycling, degradation and insertion of newly synthesized receptors are all contribute to the apparent rate of down-regulation in these measurements. Therefore, whether manipulations with IRS1 levels affect the AP-2 mediated CME of the receptor is not directly demonstrated. Notably, receptors are shown to accumulate in large clathrin structures on the cell-bottom membrane. These clathrin plaques are "endocytosis-passive" in many types of cells. Localization in these structures may not be as a "positive" endocytic event and is not a useful correlate of the internalization rate

We agree with the issues raised by the reviewer. In the original manuscript, we have shown that phospho-IGF-IR at the cell surface started to decline 3 hours after IGF-I stimulation (Figure 2A in the original manuscript). Now we show that this was overestimated due to the contribution of recycling. We have clearly observed that in the presence of primaquine surface IGF-IR as well as phospho-IGF-IR started to reduce within 30–60 minutes after IGF-I treatment (Figure 3C in the revised manuscript). Therefore, the kinetics of IGF-IR internalization in L6 cells seems comparable with other CME cargoes.

In surface biotinylation assay described in the original manuscript, apparent changes in surface IGF-IR may reflect other events than endocytosis, including recycling and de novo synthesis of IGF-IR, as the reviewer indicated. Indeed, during long-term stimulation of IGF-IR (3 hour), we reproducibly observed the modest increase in precursor IGF-IR (Figure 3—figure supplement 2A in the revised manuscript), indicating non-negligible contribution of newly-synthesized IGF-IR in the hour-scale assay. We used cycloheximide which inhibited the increase in precursor IGF-IR observed in long-term IGF-I-stimulated cells. IGF-I reduced surface IGF-IR in the presence of cycloheximide (Figure 3—figure supplement 2A in the revised manuscript). Combined with the experiments regarding IGF-IR recycling, our data support the existence of ligand-induced IGF-IR internalization in L6 cells.

To accurately evaluate the internalization, we chose the shorter time-course than the original manuscript to avoid the contribution of de novo IGF-IR synthesis. In addition, we performed the assays in the presence of primaquine to measure the net endocytic rate of IGF-IR after the ligand exposure. To test whether ectopic expression of IRS-1 affects AP-2-mediated internalization of IGF-IR, we used L6 cells stably expressing GFP, GFP-IRS-1 WT, or GFP-IRS-1 3YA. While surface IGF-IR levels were reduced within 1 hour after ligand stimulation in cells expressing GFP and GFP-IRS-1 3YA, such reduction turned to be significantly slower in cells expressing GFP-IRS-1 WT as revealed by surface biotinylation assay (Figure 4A in the revised manuscript). The data suggest that IGF-IR internalization is inhibited by the IRS-1 binding to AP2.

As described above, assays based on direct chasing of surface-labeled IGF-IR measure both basal and ligand-dependent internalization of IGF-IR. Therefore, we consider that measuring phospho-IGF-IR distribution by confocal microscopy is a superior way to evaluate the ligand-dependent internalization among our tools. In the original manuscript, AP2 knockdown inhibited the targeting of phospho-IGF-IR into lysosomes induced by IGF-I (Figure 3—figure supplement 1B, C in both original and revised manuscript), demonstrating that the trafficking of phospho-IGF-IR from the cell surface to lysosomes depends on AP2. Notably, phospho-IGF-IR accumulated in lysosomes in IRS-1-depleted cells 1 hour after IGF-I stimulation when phospho-IGF-IR is predominantly localized to the plasma membrane in control cells (Figure 5—figure supplement 1A, B in both original and revised manuscript), indicating that knockdown of IRS-1 accelerates the targeting of phospho-IGF-IR from cell surface to lysosomes. These data support the notion that manipulation of IRS-1 levels influences the AP2-dependent internalization of IGF-IR in response to the ligand.

The reviewer raised the important issue regarding colocalization of IGF-IR with AP2/clathrin at the cell bottom membrane. The function of clathrin plaque is still much debated. Clathrin plaque has been regarded as an endocytically inactive, long-lived structures (Batchlder et al., 2010; Grove et al., 2014), whereas it has been found to be actively internalized (Saffarian et al., 2009). In addition, a recent study using super-resolution microscopy revealed that clathrin-coated pits juxtaposed to plaques are easily misclassified as plaque on cell bottom membrane in the conventional TIRF microscopy (Leyton-Puig et al., 2017). As the reviewer indicated, it is hard to conclude that the colocalization of IGF-IR and AP2 at the TIRF field well reflects the efficacy of IGF-IR internalization because of the limited spatial resolution of our TIRF microscopy. Nevertheless, our observations demonstrate that ectopic expression of GFP-IRS-1 and inhibition of IRS-1 degradation by Torin1 resulted in the diffused localization of phospho-IGF-IR at TIRF field and reduced the colocalization with AP2, suggesting that manipulation with IRS-1 levels affects the ligand-dependent changes of surface IGF-IR distribution. We agree it would be important to clarify the nature of clathrin plaque and its engagement in IGF-IR endocytosis as the reviewer has noted, but we consider that clarification would not change the overall conclusions of the paper. To avoid the over-discussion, we have tried to make the text clearer as follows.

Abstract: We changed the phrase “clathrin-coated pits” to “clathrin-coated structures.”

Results section: We deleted the sentence describing the possible engagement of IGF-IR in CCPs incorporation.

Discussion section: We changed the phrase “CCPs” to “AP2-positive spots.” In addition, we added the description about the necessity of higher-resolution analyses to conclude the IGF-IR targeting to CCPs.

3) The authors propose that phosphotyrosine prevents the binding of the IRS-1 Yxxø motifs to AP-2. However, treatment of cells with IGF-1 does not inhibit the co-immunoprecipitation of IRS-1 with AP-2. The authors argue that this is because of low stoichiometry tyrosine phosphorylation of IRS-1. This could be tested by co-immunoprecipitation analysis.

In response to the reviewer’s suggestion, we performed the pull-down assay to evaluate the stoichiometry of tyrosine phosphorylation of IRS-1 (Figure 1—figure supplement 1D). This assay is based on the molecular feature of m2 that cannot recognize the phosphorylated YxxF sequence. If majority of IRS-1 is tyrosine-phosphorylated, the fraction of IRS-1 that can be pulled down by recombinant m2 would be reduced in the lysates of IGF-I-stimulated cells. However, our data demonstrate that IGF-I did not largely influence the amount of IRS-1 fraction that was pulled down by m2 (Figure 1—figure supplement 1E, F). In addition, we never detected phosphorylated IRS-1 in the pull-down fraction when equivalent amount of immunoprecipitated IRS-1 was simultaneously compared. Therefore, we conclude the low concentration of tyrosine-phosphorylated IRS-1 in IGF-I-stimulated cells in our observation.

4) The interpretation of the IRS-1 over-expression experiments is unclear. If IRS-1 binds AP-2, this should inhibit the internalization of many AP-2 cargos, but this was not observed – e.g. transferrin receptor. The authors argue that this could be because IRS-1 has some special localization within the cell; would this also be true for over-expressed IRS-1? If IRS-1 is an AP-2 competitor, why does it not inhibit internalization of other AP-2 dependent proteins when over-expressed? Moreover, if the competition for AP-2 is restricted to the local environment of the IGF-1R, does this mean that it does not affect other cell surface proteins that signal through IRS-1 signals (e.g. integrins) and if so, why not? Might this mechanism also affect transmembrane tyrosine phosphatases that dephosphorylate IGF-1R?

The reviewer raises an important question. In our original data, GFP-fused IRS-1 localized to submembraneous actin fibers that colocalize with a portion of AP2 as revealed by TIRF microscopy (Figure 4—figure supplement 1D). Such localization of endogenous IRS-1 has also been demonstrated by the previous study (Clark et al., 1998). Since actin cytoskeleton possesses critical roles in CME (Kaksonen et al., 2006), we speculate that IRS-1 may locally regulate the specific cargo endocytosis through association with a portion of AP2 at the actin cytoskeleton.

We have tested whether overexpression of IRS-1 affects endocytosis of other receptors than IGF-IR. Ectopic expression of IRS-1 did not change the transferrin receptor endocytosis as revealed by the fluorescent-labeled transferrin uptake. We further added the data showing that transferrin receptor does not physically interact with IGF-IR (Figure 4—figure supplement 1A in the revised manuscript). As the reviewer suggested, we analyzed endocytosis of integrin b1 and EGFR, both of which physically interact with IGF-IR as validated by co-immunoprecipitation (Figure 4—figure supplement 2A, E in the revised manuscript). The transfection of mRFP-IRS-1 partially inhibited endocytosis of integrin b1 in non-stimulated condition as assessed by uptake of anti-integrin b1 antibody (Figure 4—figure supplement 2B–D in the revised manuscript). In addition, the transfection of mRFP-IRS-1 partially inhibited endocytosis of EGFR in the early period of EGF stimulation. These observations raise the possibility that IRS-1 influences endocytosis of cargoes in the close proximity of IGF-IR.

In this revision, we could not test endocytosis of transmembrane phosphatases targeting IGF-IR due to the time limitation. PTP1B is one of the characterized phosphatases for IGF-IR, but is localized to the endoplasmic reticulum, not to plasma membrane. Although we did not rule out the potential effect of IRS-1 on localization/endocytosis of other transmembrane phosphatases, we consider that PTP1B is not suitable target for analysis in our case. We have not had time to assess other potential phosphatases targeting IGF-IR but agree it would be interesting although as the reviewer has noted, not necessary to the overall conclusions of the paper.

5) The kinetics of endocytosis of pIGF-IR and signaling are on different scales. The effects of the depletion or overexpression of IRS1 are evident on pAkt only after 6-12 hours (Figure 7) or later. The effects of IRS1 level alterations can be alternatively interpreted by the increase or decrease of the amount of phosphoIRS1 capable of engaging PI3K. The lack of the effect of the blockade of the receptor endocytosis by mu2 depletion of Akt activity (Figure 7—figure supplement 1D and 1E) is puzzling. The authors' interpretation of this data is that slow endocytosis is not sufficient in order to prolong Akt activity, and that the presence of high levels of IRS1 is also required. Such interpretation is difficult to reconcile with the model.

We thank the reviewer for the helpful questions. As the reviewer suggested, the cause for IRS-1-mediated increase in the duration of the Akt signaling was unclear in the original manuscript. We have now performed additional experiments to assess the role of IRS-1 engaging the PI3K-Akt pathway in the sustainability of downstream signaling by using the IRS-1 mutant ΔPTB. Although ectopic expression of IRS-1 ΔPTB strongly prolonged surface phospho-IGF-IR retention (Figure 2F, G in both original and revised manuscripts), it failed to prolong phospho-Akt (Figure 7—figure supplement 1F in the revised manuscript), suggesting that the ability of IRS-1 to engage PI3K is required for prolonging the Akt signaling. The interpretation of the reviewer therefore is correct. Nevertheless, comparing the consequences of phospho-IGF-IR and phospho-Akt in cells stably expressing GFP-IRS-1 WT, IRS-1 3YA, ΔPTB, and IRS-2, our data indicate that the IRS-1 ability to bind AP2 and promote surface retention of phospho-IGF-IR is also required for prolonging the Akt signaling. In addition, AP2 knockdown prolonged phospho-IGF-IR, but failed to sustain the Akt signaling, as like the events observed in IRS-1 ΔPTB-expressing cells (Figure 7—figure supplement 1D, E in the revised manuscript). In AP2-depleted cells, IRS-1 degradation, a consequence of negative feedback, was normally induced after long-term IGF-I treatment, indicating that the IRS-1 function to activate PI3K is shut down in the later period. Collectively, the effect of IRS-1 on prolonging the Akt signaling is likely based on two independent functions of IRS-1: the first is the binding to AP2, and the second is the activating PI3K via its tyrosine phosphorylation. We have tried to make clear the discussion of this issue and focused upon the relationship between two independent functions of IRS-1 in prolonging downstream signaling in Discussion section.